# Development and evaluation of a high-resolution reanalysis of the East Australian Current region using the Regional Ocean Modelling System (ROMS 3.4) and Incremental Strong-Constraint 4-Dimensional Variational data assimilation (IS4D-Var)

Colette Kerry[1], Brian Powell[2], Moninya Roughan[1], and Peter Oke[3]

[1]University of New South Wales, Sydney, NSW, 2052, Australia
[2]University of Hawai'i at Manoa, Honolulu, Hawaii, USA
[3]CSIRO Marine and Atmospheric Research, Hobart, Australia

*Correspondence to:* Colette Kerry (c.kerry@unsw.edu.au)

**Abstract.**

As with other Western Boundary Currents globally, the East Australian Current (EAC) is highly variable making it a challenge to model and predict. For the EAC region, we combine a high-resolution state-of-the-art numerical ocean model with a variety of traditional and newly available observations using an advanced variational data assimilation scheme. The numerical model is configured using the Regional Ocean Modelling System (ROMS 3.4) and takes boundary forcing from the BlueLink ReANalysis (BRAN3). For the data assimilation we use an Incremental Strong-Constraint 4-Dimensional Variational (IS4D-Var) scheme, which uses the model dynamics to perturb the initial conditions, atmospheric forcing, and boundary conditions, such that the modelled ocean state better fits and is in balance with the observations. This paper describes the data assimilative model configuration that achieves a significant reduction of the difference between the modelled solution and the observations to give a dynamically-consistent 'best-estimate' of the ocean state over a 2-year period. The reanalysis is shown to represent both assimilated and non-assimilated observations well. It achieves mean spatially-averaged RMS residuals with the observations of 7.6cm for SSH and $0.4°$ C for SST over the assimilation period. The time-mean RMS residual for subsurface temperature measured by Argo floats is a maximum of $0.9°$ C between water depths of 100-300m and smaller throughout the rest of the water column. Velocities at several offshore and continental shelf moorings are well represented in the reanalysis with complex correlations between 0.8-1 for all observations in the upper 500m. Surface radial velocities from a high-frequency radar array are assimilated and the reanalysis provides surface velocity estimates with complex correlations with observed velocities of 0.8-1 across the radar footprint. Comparison with independent (non-assimilated) shipboard CTD cast observations shows a marked improvement in the representation of the subsurface ocean in the reanalysis, with the RMS residual in potential density reduced to about half of the residual with the free-running model in the upper eddy-influenced part of the water column. This shows that information is successfully propagated from observed variables to unobserved regions as the assimilation system uses the model dynamics to adjust the model state estimate. This is the first study to generate a reanalysis of the region at such a high resolution, making use of an unprecedented observational data set and using an assimilation method that uses the time-evolving model physics to adjust the model in a dynamically consistent way. As such, the reanalysis potentially represents

a marked improvement in our ability to capture important circulation dynamics in the EAC. The reanalysis is being used to study EAC dynamics, observation impact in state-estimation and as forcing for a variety of downscaling studies.

# 1  Introduction

The East Australian Current (EAC) is the Western Boundary Current (WBC) of the South Pacific subtropical gyre, flowing poleward along the east coast of Australia. The EAC has the weakest mean flow of the WBCs associated with the subtropical gyres (Mata et al., 2000) but its flow is characterised by high eddy variability (Mata et al., 2006) comparable with stronger WBCs such as the Gulf Stream, the Kuroshio and the Agulhas Current (e.g. Gordon et al. (1983); Feron (1995)). The EAC forms in the South Coral Sea (15-24° S) and intensifies as it flows along the coast of southeast Queensland and northern New South Wales (NSW) (22-35° S) – refer to Figure 1 for geographical location. The current strengthens as the continental shelf narrows to 15 km at its narrowest point (31° S) and typically separates from the coast between 31-33° S (Cetina Heredia et al., 2014). Turning eastward to form the Tasman Front, the current sheds large warm- and cold-core eddies in the Tasman Sea every 90-110 days (Oke and Middleton, 2000; Cetina Heredia et al., 2014). The high eddy variability makes the EAC a challenging system to observe and predict.

In general, the kinetic energy of the ocean is dominated by submesoscale and mesoscale eddies that fluctuate on time scales of days to months and on spatial scales of tens to 100s of kilometers and exceeds the mean flow by an order of magnitude or more (Stammer, 1997; Ferrari and Wunsch, 2009). Eddies are typically generated by barotropic or baroclinic instabilities which are difficult to forecast so effective state-estimation and prediction of the mesoscale circulation requires data assimilation techniques that combine ocean observations with a dynamical model. Much of the effort towards data assimilative modelling of the EAC region has been as part of the development of Australia's Bluelink Ocean Data Assimilation System (BODAS) (Oke et al., 2005, 2008, 2013) which uses an Ensemble Optimal Interpolation (EnOI) based scheme. The BODAS has been a useful and relatively efficient method to provide ocean state estimates of the Australian region. EnOI uses long-run statistics to generate the covariance of model points and assimilates observations at a single time to generate adjusted initial conditions for each assimilation window. O'Kane et al. (2011) present an ensemble prediction study of the EAC region and identify regions of instability associated with the Tasman Front and EAC extension.

In this work we use Incremental Strong-constraint 4-Dimensional Variational data assimilation (IS4D-Var), which generates increments to adjust the model initial conditions, boundary and surface forcings such that the difference between the model solution of the time-evolving flow and all available observations is minimised over an assimilation interval. The 4D-Var scheme uses the linearised model equations and their adjoint to compute the increment adjustments, such that the model is adjusted in a dynamically consistent way to minimise the difference between the observations and the modelled time-evolving ocean state. Using the linearised equations allows dynamical connections between state variables to propagate information from observed variables to unobserved, dynamically-linked variables. Because the linearised version of the governing equations is used, rather than the full nonlinear version, the assimilation interval length is limited such that the linear assumption remains reasonably valid and the nonlinearities do not grow too large. The state estimate is a solution of the model equations, and the minimistaion

process can be used to understand the sensitivity of the modelled ocean circulation to initial conditions, boundary and surface forcing, and model parameters (e.g. Moore et al. (2009); Powell et al. (2012)). Zavala-Garay et al. (2012) used 4D-Var with ROMS to assimilate Sea Surface Height (SSH), Sea Surface Temperature (SST) and Expendable Bathythermograph (XBT) observations into a coarse resolution (18-30km) model of the EAC region. They use an empirical relationship between surface and subsurface properties to help propagate the dominant surface observations to the subsurface and improve their subsurface estimates.

Combining a state-of-the-art numerical ocean model with a variety of traditional and newly available observations, we generate a high resolution ocean state estimate of the EAC region over a 2-year period (Jan 2012 - Dec 2013). This paper describes the development and evaluation of the data assimilative model configuration. We begin by configuring a numerical model of the EAC region that is capable of representing the mean ocean circulation and its eddy variability. The model is configured to resolve the continental shelf which is 15km wide at its narrowest point and may be important in accelerating the EAC and driving the current's separation (Oke and Middleton, 2000). In order to correctly represent the spatial and temporal evolution of the eddy field, we need to constrain the model with observations. We configure a 4D-Var data assimilation system that reduces the difference between the model solution and observations, given prior assumptions of the uncertainties in the observations and the model background state. In addition to the traditional data streams (satellite derived SSH and SST, Argo profiling floats and XBT lines) we exploit newly available observations that were collected as part of Australia's Integrated Marine Observing System (IMOS, www.imos.org.au). These include velocity and hydrographic observations from a deepwater mooring array and several moorings on the continental shelf, high-frequency radar observations, and ocean gliders.

We show that the assimilation configuration developed in this work results in significant reduction of the differences between the modelled solution and the observations. As such, the reanalysis provides us with a 'best estimate' of the ocean state that is dynamically consistent within each assimilation time window. The reanalysis is being used to study the variability and separation dynamics of the EAC. Furthermore, the 4D-Var method allows us to use the reanalysis to quantify the impact of particular data streams on circulation estimates, which has the potential to provide important information for assessing and improving the observing system design. The product is also being used as boundary forcing for a variety of downscaling studies in coastal southeastern Australia. This data assimilative model represents a significant improvement on previous modelling work in the EAC for these purposes e.g. Roughan et al. (2003) which was based on climatology, Macdonald et al. (2013, 2016) which focused on process studies of warm core and cold core eddies, and Zavala-Garay et al. (2012) which used 4D-Var with a much coarser resolution.

The reanalysis development and evaluation is presented as follows. In Section 2, we describe the numerical model configuration, including validation of a 10-year free-running simulation to provide confidence that the model is correctly representing the region's circulation dynamics. In Section 3, we describe the development of the reanalysis, including the data assimilation scheme used, the assimilation configuration and the observations. The reanalysis performance is evaluated in Section 4 using a variety of metrics to illustrate the system's skill. A summary and conclusions are presented in Section 5.

## 2 Numerical Model

### 2.1 Model Configuration

We use the Regional Ocean Modeling System (ROMS, version 3.4) to simulate the atmospherically-forced eddying ocean circulation off the south eastern coast of Australia. ROMS is a free-surface, hydrostatic, primitive equation ocean model solved on a curvilinear grid with a terrain-following vertical coordinate system (Shchepetkin and McWilliams, 2005). For computational efficiency, ROMS uses a split-explicit time-stepping scheme allowing the barotropic solution to be computed at a much smaller time step than is used for the (slow-mode) baroclinic equations, using a temporal averaging filter to ensure preservation of tracers and momentum and minimise aliasing of unresolved barotropic signals into the baroclinic motions (Shchepetkin and McWilliams, 2005). The ROMS computational kernel is further described in Shchepetkin and McWilliams (1998, 2003).

Sub-grid scale horizontal mixing of momentum and tracers uses a harmonic (3-point stencil) mixing operator (Haidvogel and Beckmann, 1999), and the viscosity is derived from the horizontal divergence of the deviatory stress tensor (Wajsowicz, 1993). The diffusion and viscosity coefficients are scaled by grid-size such that less explicit diffusion occurs in the high-resolution region than in the lower resolution region. The Mellor and Yamada (1982) level-2.5, second-moment turbulence closure scheme (MY2.5) is used in parameterising vertical turbulent mixing of momentum and tracers.

The model domain (shown in Figure 1) extends from Fraser Island in the north (25.25° S) to below the NSW/Victoria border in the south (41.55° S) and nearly 1000 km offshore. The northern boundary is chosen at a latitude where the EAC remains fairly coherent and is upstream of the region of elevated eddy variability (refer to Figure 2a). The grid is rotated 20° clockwise such that it is orientated predominantly along-shore in the y-dimension and cross-shore in the x-dimension. The model has a variable horizontal resolution in the cross-shore direction, with 2.5 km (1/44°) over the continental shelf and slope that gradually increases to 6 km (1/18°) in the open ocean. The horizontal resolution is 5 km (1/22°) in the along-shore direction. The model is configured with 30 vertical s-layers distributed with a higher resolution in the upper 500m to resolve the wind driven mesoscale circulation and near the bottom for improved resolution of the bottom boundary layer. The vertical stretching scheme of Souza et al. (2014) is used, which ensures a constant-depth surface layer to better represent satellite-derived SST, better resolve the ocean surface currents, and reduce the representation error of radio-measured surface currents. The bathymetry for the model was obtained from the 50m Multibeam Dataset for Australia from Geoscience Australia (Whiteway, 2009).

In models using terrain-following coordinate systems, steep topographic gradients generate numerical errors associated with the computation of the pressure gradient term resulting in artificial along-slope flows (Haney, 1991; Mellor et al., 1994). These errors depend on the topographic steepness and the intensity of the stratification (Haidvogel et al., 2000). The variable cross-shore resolution improves the bathymetric resolution over the continental shelf and minimises pressure gradient errors over the steep topography of the continental slope, while reducing computational expense by allowing coarser resolution in the deep ocean. ROMS is effective at minimising these horizontal pressure gradient (HPG) errors (Shchepetkin and McWilliams, 2003); although, a certain degree of topographic smoothing is usually still desirable. For this study, a smoothing method has been applied in which a high priority is placed on maintaining the width of the continental shelf and preserving the seamounts that

potentially play a role in steering of the EAC, while minimising HPG errors to an acceptable level. Accurate representation of the continental shelf was considered paramount as the shelf is thought to have an important influence on the EAC (e.g. Oke and Middleton (2000)).

The model uses initial conditions and boundary forcing from the BlueLink ReANalysis version 3p5 (BRAN3) (Oke et al., 2013). BRAN is a multi-year integration of the Ocean Forecasting Australian Model (OFAM) and the Bluelink Ocean Data Assimilation System (BODAS; Oke et al. (2008)). The boundary forcing is applied daily. The Chapman condition (Chapman, 1985) is applied to the free surface and the Flather condition (Flather, 1976) is applied to the barotropic velocity so that barotropic energy is effectively transmitted out of the domain. For the free-running model, the baroclinic southern boundary conditions are clamped to the BRAN3 boundary conditions to ensure accurate representation of the outflow to the south of the domain. Radiation conditions are applied to the east and west boundaries. For the assimilation, the baroclinic boundary conditions at all three ocean boundaries are clamped to the BRAN3 boundary conditions. Baroclinic energy that does not match the BRAN3 condition is absorbed at the boundaries using a flow relaxation scheme involving a sponge layer over which viscosity and diffusivity are increased linearly by a factor of 10 from the values applied within the model domain for the northern and eastern boundaries, and a factor of 20 for the southern boundary. The size of the sponge layer is 12 grid cells (approximately 60km). Because the BRAN3 system is run with different atmospheric forcing than we use, a correction was applied to the surface heat flux forcing such that the SST from BRAN3 is in balance with the atmospheric surface forcing for each month. This correction is applied so that the surface heat flux applied through the atmospheric forcing is in balance with BRAN3, which is providing the open boundary forcing.

We begin by configuring a 10-year free-running simulation (hereafter referred to as the '10yr free run') to ensure that the model is capable of representing the mean ocean circulation and its variability. The 10yr free run is also used to provide estimates of background variability to compute background error covariances for the assimilation scheme, and the 10-year mean Sea Surface Height (SSH) field is used for addition of Sea Level Anomaly (SLA) observations for assimilation into the model. For the 10yr free run we use atmospheric forcing from the National Center for Environmental Prediction's (NCEP) reanalysis atmospheric model (Kistler et al., 2001). The atmospheric forcing fields are specified every 6 hours and used to compute the surface wind stress and surface net heat and freshwater fluxes using the bulk flux parameterisation of Fairall et al. (1996). A higher resolution atmospheric product was available for the 2-year reanalysis period. Atmospheric forcing for the 2-year model used to develop the reanalysis is provided by the 12km resolution Bureau of Meteorology (BOM) Australian Community Climate and Earth-System Simulation (ACCESS) analysis (Puri et al., 2013) and the forcing fields are specified every 6 hours.

## 2.2 Consistency of Free-running Model

The 10yr free run is performed from 2002–2011 as this is the most recent period over which BRAN3 data were available at the time of model development for use as initial and boundary forcing (BRAN3 more recently became available for the reanalysis period, 2012-2013). Comparison of the 10yr free run with observations provides validation of the ability of the model to represent the ocean dynamics in the region. The model reproduces well the spatial patterns of the time-mean and

variability of the mesoscale SSH; however, it is not expected to be in phase with the observations (e.g., the time and location of mesoscale eddies do not match). Figure 2 shows that the mesoscale SSH variability is well represented in the model compared to satellite-derived SSH data from AVISO over the 10-year period. This region of elevated SSH variability is consistent with the regions of enhanced eddy amplitude and rotational speed shown in Everett et al. (2012).

Mean cross-shore sections of alongshore velocity and temperature for the 10-year modelled period reveal a southward flowing EAC and the associated upslope thermocline tilt (Figure 3, top and middle panels). The sections shown cross the coast near Brisbane, where the EAC is found to be most coherent (27.5° S), at Coffs Harbour, just upstream of the typical EAC separation zone (30.3° S) and at Sydney, downstream of the EAC separation zone (33.9° S) (Figure 2b). The mean temperature sections compare very well with the corresponding mean temperature sections from BRAN3 over the 10-year
period, as expected as the ROMS model receives its boundary forcing from BRAN3. The mean temperature sections also match the corresponding mean temperature sections from the CSIRO Atlas of Regional Seas climatology well (CARS, Ridgway et al. (2002)), shown in the bottom panel of Figure 3. There are some small differences which are not surprising given that the CARS data covers a longer averaging period and is mapped at a much courser horizontal resolution (0.5°). In particular, difference plots (not shown) reveal some differences over the continental shelf and slope, which is not well resolved by CARS. The
comparison provides confidence that our ROMS model is representing the mean thermal structure of the ocean well.

    Alongshore transport through the same three cross-shore sections for the full water column is computed daily and the mean, standard deviation, minimum and maximum transports are shown in Table 1. Mean transport at Coffs Harbour is greater than upstream at 27.5° S due to recirculation, as described by (Ridgway and Hill, 2009). The EAC typically separates from the coast south of Coffs Harbour and mean transport through the Sydney cross section is approximately one third of the transport
at Coffs Harbour. This is consistent with Ridgway and Godfrey (1997), who estimate that about a third of the current's transport continues southward of the separation zone. Mata et al. (2000) compute transport from a mooring array located at 30° S, from the coast out the 154.4° E (a similar section to our Coffs Harbour section) between September 1991 and March 1994. They find a mean total transport of 22.1 Sv southward with an Root Mean Squared (RMS) variability of 30 Sv. This compares well with the transport through the Coffs Harbour section from the 10yr free run, which has a mean of 21.9 Sv poleward and a standard
deviation of 31.7 Sv.

    The model configuration is capable of producing the mean dynamical features of the EAC and representing the SSH variability. Thus, using 4D-Var data assimilation we aim to constrain the model with two years of observational data to examine the evolution of the EAC during this period.

## 3   Reanalysis Development

### 3.1   Configuration

The reanalysis is configured for the 2-year period of 2012–2013 because of the availability of significant observational resources during this time; in particular, a mooring array deployed to capture the transport of the EAC (as detailed in the next section). The reanalysis model uses initial conditions and boundary forcing from BRAN3 and atmospheric forcing provided by

the 12km resolution BOM ACCESS analysis, which was not available over the 10yr free run testing period described above. The simulation is spun-up over a 1-month period before we begin assimilation on Jan 1 2012. A surface heat flux correction was applied such that the new atmospheric surface forcing is in balance with SST from BRAN3 for each month. To ensure that the higher-resolution atmospheric forcing did not significantly alter the previous model comparison, we integrated the model

for two years without assimilation (hereafter referred to as the '2yr free run') and compared the model-derived sea surface temperatures (SST) with those from the advanced very-high resolution radiometer (AVHRR) satellite data. The 2yr free run model and SST observations exhibit no net bias over the 2012-2013 period (not shown). In addition, the temperature-salinity (T-S) diagram of data from Argo floats over the 2-year reanalysis period matches well with the corresponding T-S diagram for the 2yr free run output interpolated onto the Argo float locations and times (Figure 4). This provides further confidence that

the ROMS model configuration is capable of simulating the vertical structure of temperature and salinity in the region.

## 3.2   Data Assimilation Scheme

To generate the full reanalysis, we combine the model with the observations in a way that uses the model physics to compute increments in initial conditions, boundary and surface forcing to generate a state estimate that better fits the observations. In this regard, we are looking for the model to represent the observations, not replicate the observations. If the model is capable of

representing all of the observations in time and space using the physics of the model, then we should have the most complete description of the ocean-state available. To accomplish this, we use incremental strong-constraint four-dimensional variational assimilation (IS4D-Var). IS4D-Var uses variational calculus to solve for increments in model initial conditions, boundary conditions and forcing such that the difference between the modelled solution and all available observations is minimised – in a least-squares sense – over the assimilation window. This is achieved by minimising an objective cost function, $J$, that

measures normalised deviations of the modelled ocean state from the observations as well as from the modelled background state (the model prior).

The forward integration of the nonlinear model equations, given a prior estimate of the initial conditions, surface and boundary forcings, provides an estimate of the background state. The evolution of the state vector, $\mathbf{x}$, for times $t = t_1, ....., t_{i-1}, t_i, ....., t_n$ can be written as

$$\mathbf{x}(t_i) = \mathcal{M}(t_i, t_{i-1})(\mathbf{x}(t_{i-1}), \mathbf{f}(t_i), \mathbf{b}(t_i)), \qquad (1)$$

where $\mathcal{M}$ represents the nonlinear model equations operating on $\mathbf{x}(t_{i-1})$ and subject to forcing $\mathbf{f}(t_i)$ and boundary conditions $\mathbf{b}(t_i)$. The initial time for each data assimilation cycle is denoted by $t_0$ and the model time step is $t_i - t_{i-1}$. The goal of the assimilation system is to generate a vector of increments that are added to the model initial conditions, boundary conditions and forcing such that the quadratic cost function, $J$, is minimised. The increments describe departures of the initial conditions,

surface forcing, and open boundary conditions from those applied to the model prior, such that,

$$\mathbf{x}(t_0) = \mathbf{x}^b(t_0) + \delta\mathbf{x}(t_0) \qquad (2)$$

$$\mathbf{f}(t_i) = \mathbf{f}^b(t_i) + \delta\mathbf{f}(t_i) \tag{3}$$

$$\mathbf{b}(t_i) = \mathbf{b}^b(t_i) + \delta\mathbf{b}(t_i) \tag{4}$$

where $\mathbf{x}^b(t_0)$ represents the background circulation initial conditions and $\mathbf{f}^b(t_i)$ and $\mathbf{b}^b(t_i)$ are the background circulation surface forcing and boundary conditions at $t = t_i$, respectively. Because the increments $\delta\mathbf{x}$, $\delta\mathbf{f}$ and $\delta\mathbf{b}$ are assumed to be small relative to the background fields, they can be approximately described by the linearised model equations, referred to as the tangent linear model. Utilising Bayesian inference and assuming Gaussian uncertainties in the observations and model prior, we can formulate a cost function that is a function of the increment adjustments. Following Courtier (1997), we define the increment vector

$$\delta\mathbf{z} = (\delta\mathbf{x}(t_0)^T, \delta\mathbf{f}^T(t_1), ...., \delta\mathbf{f}^T(t_n), \delta\mathbf{b}^T(t_1), ...., \delta\mathbf{b}^T(t_n))^T \tag{5}$$

representing the increments to the initial conditions (time $t_0$), and the surface forcing and boundary conditions for model times $t_1$ to $t_n$. The cost function can then be written as

$$J(\delta\mathbf{z}) = \frac{1}{2}\sum_{i=0}^{n}(\mathbf{H}_i\mathbf{M}(t_i,t_0)\delta\mathbf{z} - \mathbf{d}_i)^T\mathbf{R}_i^{-1}(\mathbf{H}_i\mathbf{M}(t_i,t_0)\delta\mathbf{z} - \mathbf{d}_i) + \frac{1}{2}(\delta\mathbf{z})^T\mathbf{P}^{-1}(\delta\mathbf{z}) = J_o + J_b \tag{6}$$

where $\mathbf{M}(t_i,t_0)$ represents the tangent linear version of the nonlinear model equations $\mathcal{M}$, integrated from $t_0$ to $t_i$. The difference between the modelled background state and the observations is represented by the innovation vector, given at each time $t_i$ by $\mathbf{d}_i = \mathbf{y}_i - \mathbf{H}_i(\mathbf{x}^b(t_i))$; where $\mathbf{y}$ are the observations and $\mathbf{H}_i$ is the linear operator that interpolates the background circulation to observation points in space and time. $\mathbf{R}$ is the observation error covariance matrix and $\mathbf{P}$ is the background error covariance matrix. The observation term of the cost function, $J_o$, represents the difference between the model and the observations, and is obtained by the squared difference between the observations and the model given the integration of the increment adjustment through the tangent linear model, weighted by the inverse of the observation error covariance. $J_b$ is given by the squared increment, weighted by the inverse of the background error covariance matrix which describes the uncertainty in the initial conditions, surface and boundary forcing.

The first step of the assimilation procedure is the forward integration of the nonlinear model equations to estimate the background state (referred to as the first *outer* loop of the assimilation methodology), from which the initial cost function is computed. We seek to minimise the cost function by equating the gradient to zero. The gradient of the cost function is given by

$$\nabla_{\delta z}J = \sum_{i=0}^{n}\mathbf{M}(t_i,t_0)^T\mathbf{H}_i^T\mathbf{R}_i^{-1}(\mathbf{H}_i\mathbf{M}(t_i,t_0)\delta\mathbf{z} - \mathbf{d}_i) + \mathbf{P}^{-1}(\delta\mathbf{z}), \tag{7}$$

where $\mathbf{M}(t_i, t_0)^T$ is the adjoint of the tangent linear model equations. To compute the graident, the tangent linear model is integrated using the increment $\delta\mathbf{z}$ (for the first iteration, $\delta\mathbf{z} = 0$) and $\mathbf{H}_i\mathbf{M}(t_i, t_0)\delta\mathbf{z} - \mathbf{d}_i$ is computed. The adjoint model is then used to compute the first term of Equation 7 and $\nabla_{\delta x}J$ is computed. A Lanczos-based conjugate gradient method is used to determine how far to step in the direction of the gradient to reduce $J$ and a new increment, $\delta\mathbf{z}$, is generated. Subsequent integrations of the tangent linear and adjoint model (referred to as the *inner* loops) are continued to generate subsequent increments to minimise $J$. In practice, the *inner* loops can be continued until $J$ is reduced by a certain ratio or, as in this study, a set number of *inner* loops can be completed that are found to give an acceptable reduction in $J$. We do not find the true minimum of $J$, but rather an acceptable reduction. After the last *inner* loop, the final increment is applied to generate new initial conditions, boundary and surface forcing. The new integration of the nonlinear model, given the increment adjustments, completes the *outer* loop. The final *outer* loop provides the 'best estimate' of the ocean state (the analysis) which is constrained to satisfy the nonlinear model equations (strong-constraint) and better represent the observations over the assimilation window. The analysis provides an improved estimate of the initial state for the subsequent assimilation window.

An advantage of this assimilation method is that it makes use of the dynamical connections between the model fields, such that observed variables propagate information to unobserved, dynamically-linked variables. Because the linearised model equations are used for the cost function minimisation, the length of the assimilation window is limited by the time over which the tangent linear assumption remains reasonable. For a thorough description of the IS4D-Var formulation, the reader is referred to Moore et al. (2011c). The ROMS 4D-Var implementation is well described by (Moore et al., 2011c, a, b), and it has been used successfully in ROMS applications (e.g., Di Lorenzo et al. (2007); Powell et al. (2008); Powell and Moore (2008); Broquet et al. (2009); Matthews et al. (2012); Zavala-Garay et al. (2012); Janeković et al. (2013); Souza et al. (2014)).

### 3.3 Assimilation Configuration

The goal of the assimilation is to combine an uncertain model with uncertain observations to generate a circulation estimate that has reduced uncertainty and better represents the observations. To do this we solve for the nonlinear ocean solution that is dynamically consistent with the observations and is free within the uncertainties in the system. As such, specification of the prior model and observation uncertainties is important. These uncertainties are prescribed in the background and observation error covariance matrices, respectively, and are important scaling factors in the cost function, $J$ (refer to Equation 6). The specification of the background and observation error covariances is described in Sections 3.4 and 3.5 below, and their consistencies checked in Section 4.1.

The minimisation of $J$ is performed over a specified time window in a sequence of linear least-squares minimisations in the *inner* loops, and the nonlinear model trajectory is updated in the *outer* loops. Through experimentation, we found that one *outer* loop, with 14 *inner* loops, gives an acceptable reduction in $J$ for a reasonable computational cost. Cost function convergence is shown in Section 4.2. We aim for the longest time window available without nonlinearities growing too large. Linearity experiments (not shown) indicated that for this model configuration, the linear assumption remains acceptable for typical perturbations over 5 days, so we chose that as our window-size. We overlap the 5-day assimilation windows by one-day,

such that each subsequent assimilation cycle is initialised 4 days after the start of the previous 5-day cycle. The overlap allows us to produce a blended product which is constructed as a post processing step using a weighted average of the overlapping times from adjacent assimilation windows to build a continuous signal. The blended product can then be used for dynamical analysis and further nesting.

The ROMS 4D-Var allows for controlling both the initial conditions and the time-varying atmospheric and boundary forcing. We adjust the atmospheric forcing every 3 hours and the open boundary conditions every 12 hours. The heat flux is the dominant adjustment in the atmospheric forcing over most of the domain, with the wind adjustment dominating in the vicinity of the HF radar.

## 3.4   Observations and Observation Prior Uncertainties

The reanalysis time period (2012–2013) was chosen because it contains the greatest number of available observations, including a full depth mooring array that resolves the EAC transport, that was deployed from 1 Apr 2012 to 26 Aug 2013. Other available subsurface observations and satellite-derived surface observations are also sourced for this time period. Figure 5a shows the location of Argo profiling float observations, coloured by time of occurrence, and Figure 5b shows the location of all other observations, with the exception of the satellite-derived SSH, SST and Sea Surface Salinity (SSS). The number of processed

observations assimilated for each 5-day assimilation window is shown in Figure 6a, with a break-down of the provenance of the temperature observations in Figure 6b. The observations and their respective processing for assimilation into the reanalysis are detailed in the subsections below.

      The uncertainties in the observations are specified to prevent 'over-fitting' the solution to uncertain observations. The observation uncertainty is a combination of the uncertainty in the observation itself and just as significantly, the uncertainty in the

model's ability to represent that observation (referred to as representation error). The observational uncertainties are prescribed in the observation error covariance matrix, $\mathbf{R}$, which is an diagonal $N$ by $N$ matrix, where $N$ is the number of observations. The representation errors depend on the spatial and temporal resolution of the model and the processes resolved. Observations that capture processes unresolved by the model must be either filtered or an uncertainty applied that accounts for the unresolved process. If multiple observations from the same instrument exist in the same horizontal and vertical grid cell taken

within the same model time-step (5 minutes), the observations are averaged and this value is assimilated within that grid cell at the appropriate time. The variance of those observations provides a lower-bound to the representation error as the model can express only a single value. Oftentimes, this error is greatest where the model resolution (and therefore its physics) cannot represent finer-scale dynamics captured in the observations. A thorough discussion of representation error can be found in Oke and Sakov (2008).

We describe the observations used in the section below, and detail the observation uncertainties specified for each. The consistency of these uncertainty estimates is checked in Section 4.1.

### 3.4.1 Satellite-derived Sea Surface Height

Archiving, Validation and Interpretation of Satellite Oceanographic Data (AVISO), France, produce global, daily, gridded ($1/4°$ x $1/4°$) mean sea level anomaly (SLA) data produced by merging of all available along-track satellite altimetry data, computed with respect to a seven-year mean. The AVISO data provides a daily statistical field giving a synoptic view of the SSH. The

AVISO SLA data is added to the dynamic SSH mean from the 10yr free run described above to generate sea level data for assimilation that is consistent with the ROMS model bathymetry and configuration. We prescribe an observation uncertainty of 6cm. The error in the AVISO delayed-time global SLA product due to noise for the region is estimated at 2cm (CNES, 2015). We include a further 4cm of uncertainty because, in this case, the model resolves far more structure at smaller spatial scales than is capable in the observations. The AVISO fields provide a statistical fit to along-track SSH data and the observation

uncertainty allows for imbalances between this statistical field and a dynamically balanced SSH field required by the model. We exclude SSH observations that were taken over water depths less than 1000m. This is because the observations are noisy on the continental shelf and the AVISO gridded product is not able to resolve the processes that occur here.

The gridded AVISO product is used to constrain SSH, rather than the along-track altimetry, to ensure that the constraint is projected into the baroclinic ocean state solution. The use of along-track SSH data successfully with 4D-Var relies on the

prescription of balanced terms in the background error covariance matrix to describe the covariance between SSH and the subsurface ocean (refer to Section 3.5). This is a topic of further research.

### 3.4.2 Satellite-derived Sea Surface Temperature

We use SST from the US Naval Oceanographic Office's Global Area Coverage Advanced Very High Resolution Radiometer level-2 product (NAVOCEANO's GAC AVHRR L2P SST). The product does not provide observations through clouds

but contains useful observations close to the coast. Data is available 2-3 times per day. A product error is specified in the NAVOOCEANO SST product (Andreu-Burillo et al., 2010), with an error for each data point of $0.38-0.4°$ C. As the resolution of the data is similar to the resolution of the model, the observation uncertainty for the assimilation is chosen to be equal to this product error.

### 3.4.3 Satellite-derived Sea Surface Salinity

Sea Surface Salinity (SSS) has been observed from space for the first time by the National Aeronautics and Space Administrations's (NASA) Aquarius satellite (www.aquarius.umaine.edu/). We make use of the Level 3 gridded salinity product which provides daily fields at a $1°$ resolution. The observation uncertainty is set to 0.4. There is a product error of around 0.2 for the Aquarius SSS data and 0.4 is chosen to account for additional uncertainty due to processes not resolved by the observations or the model. The value is considerably higher than the uncertainties specified for other *in-situ* salinity observations. Similarly to

SSH, any data taken over water depths less than 1000m depth were eliminated.

### 3.4.4 Argo floats

Argo is an international program consisting of nearly 4,000 free-drifting profiling floats that measure the temperature and salinity of the upper 2000m of the global ocean (www.argo.ucsd.edu). The Argo float locations in our model domain for 2011-2012, and the times at which they occur at those locations, are shown in Figure 5a. The Argo data points are averaged to the model grid and 5-minute time-step.

Uncertainty profiles are defined to specify the nominal minimal uncertainties for subsurface temperature and salinity. To devise the profile shapes, temperature and salinity variance is computed for each month of the year from the 10-yr free run. The monthly variances are spatially averaged over the model domain and averaged in time to give a single variance profile for both temperature and salinity. The profiles are then scaled to provide variance profiles appropriate for the nominal minimum observation error variance, based on preliminary assimilations and checks against the diagnostics described in Section 4.1 (computed throughout the water column). The uncertainty profiles are shown in Figure 7 (standard deviation is plotted instead of variance so the units are more intuitive for the reader). The profiles provide greater uncertainties in the depth ranges of greatest variability where representation errors are likely to be the largest. The observation error variance is specified as the maximum of this nominal minimum error variance and the variance of the observations from the same model cell.

### 3.4.5 Expendable Bathythermographs

Expendable Bathythermographs (XBT) collect temperature profiles along repeat lines sampled by merchant ships. Two transects intersect our model domain; PX34 which is the Sydney-Wellington route, and PX30 which is the Brisbane-Fiji route (only a small portion of this transect is within our model domain). Five PX30 lines took place over the assimilation period (16 Dec 2011, 8-9 Mar 2012, 13 Sep 2012, 7 Jun 2013 and 1 Nov 2013) and seven PX34 lines (3-4 Feb 2012, 23-24 May 2012, 22-23 Sep 2012, 26-27 Nov 2012, 16-18 Feb 2013, 12-13 May 2013 and 24-26 Aug 2013). The sections are sampled at 10km intervals. The XBT data points are averaged to the model grid and 5-minute time-step. The same nominal minimal uncertainty profile used for the Argo temperature observations (Figure 7) is used and the observation error variance is specified as the maximum of the nominal minimum error variance and the variance of the observations from the same model cell.

### 3.4.6 High-Frequency Radar

The Coffs Harbour high-frequency (HF) ocean radar is part of the IMOS and is managed by the Australian Coastal Ocean Radar Network (ACORN, http://imos.org.au/acorn.html). The radar is a WERA phased array system with 16-element receive arrays located at Red Rock to the north of Coffs Harbour (RRK, $29.98°$ S , $153.23°$ E) and North Nambucca to the south (NNB, $30.62°$ S , $153.011°$ E). The radars operate at a frequency of 13.920 MHz, with a bandwidth of 100 KHz and a maximum range of 100 Km.

The HF radar broadcasts and receives along defined angles in a phased-array setup and the surface current speed (towards and away from the radar site) is measured. The overlapping coverage from the two radar sites allows the surface current (u and v) vectors to be computed. Using the same assimilation procedure as detailed in Souza et al. (2014), we assimilate radial

currents, rather than the computed current velocities. The velocities have correlated errors and using the radials allows us to make use of data when only one station is available and over areas that do not overlap with the other station's measurements. We can also make use of radial data in regions where the beam intersection angle between measurements from the two stations results in high error in the velocity calculation while the radials errors are adequately low.

Radial data is available from 1 Mar 2012 to the end of the reanalysis period. The areas of HF radar coverage are shown in Figure 5b, with inset panels showing the percentage of data coverage for assimilated radials for the RRK and NNB sites in Figure 5c and 5d, respectively. Radials for each of the two stations are processed separately. At the outer range of the HF radar instrument coverage, radial values become noisy. We extract only radial values with Bragg Signal to Noise ratio >10 dB. Manual inspection of the radial values for each of the two sites was then conducted and a "good data" region was chosen for

each site every day, excluding the outer regions of coverage where noisy data is observed. Only radial data within these "good data" regions is used, and absolute radial speed values greater than $2 \, \mathrm{ms}^{-1}$ are excluded. This manual inspection was performed daily as the radii of reliable radial data varies significantly, and this method allows us to retain the maximum amount of data for assimilation. The radial speeds and angles are spatially averaged onto the model grid and a 24-hour boxcar averaging filter is used to remove tides and inertial oscillations that are not resolved by the model.

Radial speed standard error is given in the data files provided by ACORN, calculated from the mean width of the two Bragg peaks weighted by their maximum power (Wyatt, 2014). These standard errors are converted to variances and averaged as above. An error variance is then applied to each observation, given by the maximum of the averaged variances and the variance of the averaged radial speeds. The nominal minimum observation error for the surface radials is set to $0.15 \, \mathrm{ms}^{-1}$. The observation error covariance for each radial speed observation is set to the maximum of the nominal minimum observation

error covariance and the error variance computed during the averaging. Any observations where the square-root of the radial speed error variance exceeds the radial speed magnitude are removed. Radial data within one grid cell of the coast is also removed as unrealistically high values are observed here.

### 3.4.7    NSW Shelf Moorings

Data collected from 3 moorings located along the NSW continental shelf is used in this assimilation study. The moorings

collect temperature and velocity data at high sampling frequencies and are located off of Coffs Harbour, $30°$ S (CH100) and Sydney, $33.9°$ S (SYD100 and SYD140). In each case the number in the mooring name represents the approximate water depth of the mooring location. Table 2 contains details of the mooring locations and the properties observed. Temperature and velocity observations are every 8m through the water column. The data collection and quality control is described in detail in Roughan and Morris (2011).

All temperature observations taken from moorings at high sampling frequencies are low-pass filtered to remove variability at periods shorter than the inertial period (23.8 hours for Coffs Harbour and 21.5 hours for Sydney), and the observations are applied 6-hourly. For latitudes south of $30°$ S, the inertial period is less than 24 hours so the filtering does not remove the diurnal signal which may arise from internal tides and/or diurnal surface heating for near surface temperature observations. The RMS residual between the mooring temperature observations low-pass filtered at 30 hours (removing variability due to baroclinic

tides and inertial oscillations) and the observations filtered at the inertial frequency is very small compared to the nominal minimum uncertainties applied, confirming that these unresolved processes are accounted for in the observation uncertainty specification. Velocity observations are low-pass filtered at 30 hours to remove variability due to tides and inertial oscillations and applied 6-hourly. It is important to remove the tidal signal from velocity observations as the barotropic tidal velocities are of similar order of magnitude to the sub-tidal velocities.

For all observations on the continental shelf, different nominal minimum observation error variance profiles are adopted (to those used offshore for Argo and XBT) to account for increased variability due to finer scale processes that occur on the shelf that are not resolved in the model. Variance profiles for the shelf observations were computed by comparing all of the shelf observations (NSW moorings, SEQ moorings and gliders) to the 2yr free run for the 2012-2013 assimilation period to generate a nominal uncertainty profile on the shelf. Profiles were generated for all observed *in situ* variables; u and v velocity components, temperature, and salinity. u (v) uncertainty peaks at 0.12 (0.3) ms$^{-1}$ in the upper 50m reducing to 0.08 (0.1) ms$^{-1}$ at 200m depth. The shelf temperature uncertainty profile peaks at 1.2° C between 20-100m depth, reducing to 0.8° C at 200m. We doubled the computed salinity errors to give a range of 0.1-0.16 for the upper 200m on the shelf.

The observation error variance is specified as the maximum of the nominal minimum error variance and the variance from averaging observations within the same model grid cell. For velocities, the high density of the ADCP depth bins means several velocity measurements are often available for a single vertical grid layer, which can result in variances that exceed the specified nominal minimum uncertainty.

### 3.4.8 EAC Transport Array and SEQ Shelf Moorings

The EAC transport array was deployed as part of IMOS to understand the variability of the EAC, and it is comprised of five deep water moorings (EAC 1-5) which measure temperature, salinity and velocities. The array was positioned where the EAC is predicted to be most coherent and was designed to measure the mean and time-varying EAC transport (Sloyan et al., 2016). The array is continued onto the shelf slope and shelf with two moorings (SEQ400 and SEQ200) in approximate water depths of 400m and 200m, respectively. Each mooring has a suite of instruments measuring temperature, salinity, and velocities at high sampling frequencies throughout the water column. Table 2 contains details of the moorings.

All temperature and salinity observations are low-pass filtered to remove variability at periods shorter than the inertial period (26.0 hours), and the observations are applied 6-hourly. The vertical uncertainty profile used for the other off-shelf temperature and salinity observations (Figure 7) is used for the nominal minimum profile for the EAC array mooring observations. For the SEQ moorings the nominal minimum vertical uncertainty profile generated for the shelf observations was used. The velocity observations are filtered and processed in the same manner as the NSW moorings, described above. For the EAC array moorings, the nominal minimum error for u and v velocity components was specified as 0.12 ms$^{-1}$ in the upper 10m of the water column and 0.10 ms$^{-1}$ for all depths below 10m. For the SEQ moorings, the uncertainty profile generated for the shelf observations was used. Similarly to the NSW moorings, the observation error variance is specified as the maximum of the nominal minimum error variance and the variance from averaging observations within the same model grid cell.

### 3.4.9 Ocean Gliders

Autonomous ocean gliders (both SeaGliders and Slocum) were deployed as part of the IMOS by the Australian National Facility for Ocean Gliders (http://imos.org.au/anfog.html). The buoyancy controlled gliders move horizontally through the water while collecting vertical profiles of temperature and salinity. The majority of the glider missions in the model domain over the 2011–2012 time period occur on the NSW continental shelf, between $29.5°$ S and $32.3°$ S, with two missions between 25 March 2013 and 22 July 2013 extending offshore, further south and to depths of 900m (Figure 5b). Quality control flags are applied to the glider data through the IMOS processing (Australian National Facility for Ocean Gliders, 2012) and only the data deemed to be "top quality data in which no malfunctions have been identified and all real features have been verified during the quality control process" are used. The glider data points are averaged onto the model grid and 5-minute time-step. Glider temperature data in the upper 20m and salinity data in the upper 50m were removed. The uncertainty profiles computed for the shelf observations were used and the error variances for the gliders are the maximum between the variance computed from the averaging and the shelf error variance profile.

### 3.5 Model Prior Uncertainties

The background error covariance matrix, $\mathbf{P}$, should represent the expected uncertainties in the model initial conditions, surface and boundary forcings. $\mathbf{P}$ is an $M$ by $M$ matrix, where $M$ is the length of the increment vector $\delta\mathbf{z}$ (Equation 5). Because of its size $\mathbf{P}$ cannot be estimated completely or stored and we estimate $\mathbf{P}$ by factorisation, as described in Weaver and Courtier (2001), such that,

$$\mathbf{P} = \mathbf{K}_b \Sigma \Lambda L_v^{1/2} L_h L_v^{1/2} \Lambda \Sigma \mathbf{K}_b^T, \tag{8}$$

where $\mathbf{K}_b$ are the covariance operators of the balanced dynamics, $\Sigma$ and $\Lambda$ are the diagonal matrices of the background error standard deviations and normalisation factors respectively, and $L_v$ and $L_h$ are the univariate correlations in the vertical and horizontal directions. In this work, we only prescribe univariate covariance in $\mathbf{K}_b$. The dynamics are coupled through the use of the tangent linear and adjoint models in the assimilation, but not in the statistics of $\mathbf{P}$. The correlation matrices, $L_v$ and $L_h$, and the normalisation factors, $\Lambda$, are computed as solutions to diffusion equations following Weaver and Courtier (2001). The characteristic length scales chosen for $L_v$ and $L_h$ are assumed to be homogeneous and isotropic.

In the horizontal, the characteristic length scales chosen for the background error covariances are 100km for SSH, temperature and salinity and 70km for velocities. These values were chosen based on analysis of cross-correlation of SSH and complex correlation of surface velocities between points in the eddy rich Tasman Sea region from the 2yr free run. The length scale of 100km for SSH is consistent with the decorrelation scales estimated from along-track satellite data for the area by Wilkin et al. (2002) and used by Zavala-Garay et al. (2012). It is noted that shorter cross-shore length scales are likely along the coast of south-eastern Australia, as the continental shelf is narrow (15-30km) and the EAC displays a narrow jet like structure, while SSH decorrelation length scales were found to be about 100-200km in the alongshore direction by Oke and Sakov (2012).

For the vertical, semivariogram analysis of glider data on the NSW shelf by Schaeffer et al. (2015) found vertical decorrelation length scales of about 50m for both temperature and salinity on the NSW shelf. Analysis of correlations between temperature data measured by the moorings used in this study found vertical decorrelation length scales of 15-30m for the shelf moorings (NSW moorings, SEQ 200), 70m for SEQ 400 and 100-200m for the EAC deepwater array moorings (EAC 1-5). Salinity measurements were taken at SEQ200, SEQ400 and the EAC deep water array moorings and decorrelation length scales were similar to the length scales for temperature at these moorings. In the vertical, we apply characteristic length scales of 50m for salinity and 10m for temperature. The shorter length scale for temperature was adopted due to the short length scale of variability for temperature near the sea surface, as SST observations dominate. The salinity length scale is set to 50m (longer than the temperature length scale) in order to limit vertical structure in the salinity analysis increments.

Analysis of correlations between velocities measured by the moorings found vertical decorrelation length scales of 20-50m for the shelf moorings (NSW moorings, SEQ 200), 70m for SEQ 400 and 100-200m for the the the EAC deep water array moorings (EAC 1-5). Because the deep water moorings span the core of the EAC, we reduced the de-correlation length scale value to 50m in the vertical for velocity to be ensure consistency when assimilating velocities outside of the EAC and/or on the shelf.

The background error covariance matrix plays an important role in determining the spatial structure of the analysis increment and, in this oceanic region, the horizontal and vertical scales of variability differ between the mesoscale eddy field in the Tasman Sea and the smaller-scale shelf processes. Further research on the impact of applying anisotropic correlation length scales on system performance is warranted.

The background error standard deviations were estimated from the average of 5-day variances from the 10yr free run described above. These climatological variances provide an estimate of the uncertainty associated with each state variable and surface forcing field, based on the assumption that background errors are likely to be largest in regions of strong ocean variability. We choose 5-day variances as the model is nested in BRAN3 which assimilates large-scale data so we expect our model prior boundary and initial conditions to be accurate to within the typical changes to the ocean state that occur over 5-days. The same background error covariance matrix is used for each assimilation cycle.

## 4   Reanalysis Evaluation

In this section, we evaluate the performance of the assimilation procedure in terms of the consistency of the prior uncertainty assumptions, comparison with the assimilated observations, and comparison to unassimilated observations. Overall, the assimilation performs well in minimising the cost function over each assimilation interval and the corresponding reanalysis provides a good match to observations.

### 4.1   Consistency of Observation and Model Uncertainties

The analysis generated by the IS4D-Var system is dependent on the prior assumptions of the background and observation uncertainties, and the validity of these assumptions is important in determining the optimality of the analysis. A measure of the consistency of the assimilation system given the prior uncertainty assumptions can be made using a set of diagnostics

based on the innovation statistics, presented in Desroziers et al. (2005). These diagnostics are based on the observation minus background, observation minus analysis, and analysis minus background differences and provide a check of the consistency of the prior choices of the background and observation error covariances. The level of agreement between the *a priori* specified error variances ($\mathbf{P}$ and $\mathbf{R}$), and those diagnosed *a posteriori* following the methods introduced by Desroziers et al. (2005) provides a measure of the appropriateness of the estimates of $\mathbf{P}$ and $\mathbf{R}$. We find that the prior specified error variances and those diagnosed after the assimilation match well.

For SSH, square-root of the spatially-averaged diagnosed observation error variance ranges from 4.1-8.4cm with a mean value of 5.8cm, which matches the square-root of the prior observation error variance of 6cm very well. The SSH prior and diagnosed model error variances are also consistent. For subsurface temperature, the prior and diagnosed model error variances match very well. The prior observation error variances are greater than the diagnosed observation error variances for subsurface temperature; the time-mean of the square-root of the spatially-averaged prior error variances is $0.88°$ C compared to $0.48°$ C for the diagnosed errors. This prior uncertainty was necessary to account for the representation errors associated with the subsurface temperature observations. Similarly for subsurface salinity and velocities, the prior observation error variances exceed the diagnosed observation error variances. For the radial current speeds, the time-mean of the square-root of the spatially-averaged diagnosed observation error variances is $0.11\text{ms}^{-1}$, which matches the prior observation uncertainty of $0.15\text{ms}^{-1}$ well.

Another simple diagnostic to check the validity of $\mathbf{P}$ and $\mathbf{R}$ is to check the value of the cost function, $J$, at its minimum. As shown by Bennett (2002), the theoretical minimum of the cost function, for a linear system, is $N_{obs}/2$, where $N_{obs}$ is the number of observations. This minimum should be reached on each assimilation cycle if the prior background and observation error covariance estimates are correctly specified and the system is quasi-linear (Weaver et al., 2003). It is convenient to define the 'optimality' value, $\gamma = 2J/N_{obs}$, which should reach a value of $1 \pm \sqrt{2/N_{obs}}$ (Powell et al., 2008). As $N_{obs}$ is large, a value of 1 indicates correct specification of the uncertainties. The 'optimality' value provides a measure of how well the system approaches an optimal fit and, for the 5-day analysis windows over the 2-year assimilation, the values range from 0.43-1.72 with a mean value of 0.81.

Overall, the prior assumptions of observation and model background uncertainties are considered reasonable and the assimilation achieves reduced analysis uncertainty by reduction of the cost function for each assimilation interval. The cost function reduction and convergence properties are detailed in the following section.

## 4.2 Cost Function Reduction and Convergence Properties

Linear minimisation of the cost function, $J$, is performed in the *inner* loops. We use a single *outer* loop, at the end of which the final cost function is computed after the integration of the nonlinear model. Figure 8a shows the initial nonlinear cost function (black), the reduction achieved in the final (14th) *inner* loop (blue) and the final nonlinear cost function (magenta), plotted for each assimilation interval over the 2-year period. The match between the final tangent linear and the nonlinear cost functions provides confidence in the validity of the tangent-linear assumption over the 5-day assimilation window. The mean tangent-linear model cost function reduction ($1 - J_{TLM}/J_{initial}$, where $J_{TLM}$ is the cost function for the final *inner* loop) over all assimilation windows is 62%, and the mean of the subsequent non-linear model cost function reduction ($1 - J_{NLM}/J_{initial}$)

is 52%. Temperature dominates the cost function, followed by the velocities (including the radials) and SSH, with salinity playing the least dominant role. Figure 8b shows the mean cost function reduction for each of the 14 *inner* loops for all 5-day assimilation windows. The cost function reduction relative to the initial cost function increases with each *inner* loop, and the curve begins to flatten out towards the final *inner* loop showing that 14 loops is a good choice. The mean reduction in the nonlinear cost function is shown by a magenta dot in the plot and shows that minor nonlinearities persist in our assimilation windows.

## 4.3 Reanalysis Comparison to Assimilated Observations

### 4.3.1 SSH

The Root Mean Squared (RMS) observation anomaly for a particular observation location describes the variability in the observation with respect to its time-mean. This is compared to the RMS differences between the observations and the free-running model (the 2yr free run), and the observations and the analysis (i.e. the analysis error), to provide an assessment of how well the free run and the analysis match the observations relative to their typical variability. A skillful state estimate will have residuals with the observations that are much lower that the observation's typical variability.

The observation anomaly for an observed variable $v$ at a particular location is given by

$$RMS_{ObsAnom} = \sqrt{\frac{\Sigma_{t_1}^{t_n}(v(t) - \bar{v})^2}{n}}, \tag{9}$$

where $t = t_1, t_2, ....t_n$ are the observation times and $\bar{v}$ is the time-mean of the observed variable at that location. The RMS difference between the free run values (in observation space) and the observations and the analysis and observations are given by

$$RMSD_{Freerun-Obs} = \sqrt{\frac{\Sigma_{t_1}^{t_n}(v_f(t) - v_o(t))^2}{n}}, \tag{10}$$

and

$$RMSD_{Analysis-Obs} = \sqrt{\frac{\Sigma_{t_1}^{t_n}(v_a(t) - v_o(t))^2}{n}}, \tag{11}$$

respectively, where $v_o$ is the observed value, $v_f$ is the corresponding value from the free run and $v_a$ is the corresponding value from the analysis.

Figure 9a shows the RMS SSH anomaly from the observations over the 2-year assimilation period. The $RMSD_{Analysis-Obs}$ is shown in Figure 9b and shows that the SSH fields are well represented in the analyses. In Figure 9c, the domain-averaged $RMS_{ObsAnom}$, $RMSD_{Freerun-Obs}$ and $RMSD_{Analysis-Obs}$ are plotted for each 5-day assimilation window over the 2-year period, showing significant improvement in the fit to observations in the analyses. The $RMSD_{Freerun-Obs}$ is of similar magnitude to the SSH observation anomaly indicating that, as expected, the free run has no skill in predicting the timing and location

of the mesoscale eddies. The time-mean of the spatially-averaged $\text{RMSD}_{Analysis-Obs}$ over all assimilation windows is 7.6cm. This is close to the observation uncertainty for SSH of 6cm and small compared to the typical SSH variability (the time-mean of the spatially-averaged SSH observation anomalies is 23.4cm)

### 4.3.2 SST

The free-running model shows some skill in prediction of the SST due to the accuracy of the surface forcing; however, significant improvement is achieved in the analyses. The RMS SST observation anomalies describe the variability in SST over the 2-year assimilation period, including the seasonal cycle, and are shown in Figure 10a. The $\text{RMSD}_{Freerun-Obs}$ is smaller than the observation anomalies and the $\text{RMSD}_{Analysis-Obs}$ (Figure 10b) is further reduced. The time-series of spatially-averaged $RMS_{ObsAnom}$, $\text{RMSD}_{Freerun-Obs}$ and $\text{RMSD}_{Analysis-Obs}$ are shown in Figure 10c over the 2-year period. The time-mean of the spatially-averaged analysis error for all assimilation windows is 0.4° C, which is the same magnitude as the SST observation uncertainty. The free-running model and SST observations exhibit no net bias over the 2 years (as mentioned in Section 3.1), indicating that the the RMSD reduction between the free-run and the analysis is due to improved prediction of the dynamical features rather than a reduction in bias. The high variability seen in the time-series plots, particularly in the observation anomaly, is due to the patchy spatial coverage of the SST observations.

### 4.3.3 SSS

The Aquarius SSS data was included but for this assimilation configuration provides little constraint. $RMS_{ObsAnom}$ for SSS is 0.15-0.3 over most of the model domain (up to 0.5 at a few points close to the coast). The Aquarius product error itself is 0.2 and our specified observation error is 0.4, which is greater than the typical variability in SSS over most of the domain, so the assimilation does little to match the SSS observations as they are so uncertain. The $\text{RMSD}_{Freerun-Obs}$, $\text{RMSD}_{Analysis-Obs}$ and $RMS_{ObsAnom}$ are all of similar magnitude. Subsurface salinity dominates the salinity cost function as the prescribed observation uncertainties are considerably higher for SSS than the uncertainties specified for the in-situ salinity observations.

### 4.3.4 Subsurface temperature from Argo, Gliders and XBT

Subsurface observations are spatially and/or temporally sparse in comparison to satellite observations of the sea surface. The dynamical connections between surface and subsurface variables are taken into account by the adjoint and tangent-linear model such that the time-evolving model physics are used to perform the cost function minimisation. While these connections allow the surface observations to impact state estimates of the subsurface properties, subsurface observations are invaluable in improving estimates of the subsurface (e.g., Zavala-Garay et al. (2012)).

We show the improvement in subsurface temperature as measured by the Argo floats, XBTs and ocean gliders by computing the $\text{RMSD}_{Freerun-Obs}$ and $\text{RMSD}_{Analysis-Obs}$ in nominal depth bins for all observations over the model domain (Figure 11). For Argo, time-mean $\text{RMSD}_{Freerun-Obs}$ for all observations in the upper 500m of the water column is 1.7° C, reduced to 0.8° C in the analysis. For the XBT above 500m, the time-mean RMSD is reduced to 0.7° C in the analysis from 1.9° C

for the free-running model. Below 500m, the number of observations in each depth bin for Argo and XBT is too low for a meaningful comparison. The glider data mostly samples the shelf and shelf slope circulation. A great majority of the glider observations are in the upper 100m of the water column; here the time-mean $\text{RMSD}_{Freerun-Obs}$ of 2.1° C is reduced to 0.7° C in the analysis.

To investigate the relative contribution of improved representation of dynamical features and reduction in bias to the RMSD reduction between the free run and the analysis, we also compute the RMSD between the free run and the 'bias adjusted observations'. The 'bias adjusted observations' have the bias between the observations and the free run removed and, for each depth bin, are given by $v_o(t) - (\bar{v}_o - \bar{v}_f)$, where $v_o$ are the observations in the depth bin for observation times $t = t_1, t_2, ....t_n$, $v_f$ are the corresponding values from the free run and the overbar represents the time-mean of the variables over the 2-year period.

For the Argo and XBT observations, the bias between the free run and the observations is small and the $\text{RMSD}_{Freerun-Obs}$ and the RMSD between the free run and the 'bias adjusted observations' (blue and grey dashed lines in Figure 11, respectively) match closely. The RMSD reduction for the analysis (magenta line) is due to better representation of the dynamical features. The vast majority of glider observations are taken on the continental shelf in water depths less than 100m. For these shallow glider observations, the bias between the free run and the observations is approximately 1.5° C (not shown). The bias in the

analysis is close to zero and this reduction in bias contributes to the reduction in the $\text{RMSD}_{Analysis-Obs}$ compared to the free run (the RMSD between the free run and the 'bias adjusted observations' (grey dashed line) is less than the $\text{RMSD}_{Freerun-Obs}$ (blue line)). There is further reduction in the $\text{RMSD}_{Anslysis-Obs}$ (magenta line) compared to the RMSD between the free run and the 'bias adjusted observations' (grey dashed line) indicating improved representation of dynamical features. It should be noted that the glider observations below 100m represent only 2 separate glider missions (refer to Section 3.4.9), so the bias has

little meaning over this depth range.

As the Argo profiling floats measure both temperature and salinity at each observation time we are able to assess the residual reduction in terms of potential density throughout the water column (Figure 12), describing the improvement in the representation of the density structure in the analysis. The free-running model has some skill in predicting potential density as sampled by the Argo floats in the upper 500m, as the $\text{RMSD}_{Freerun-Obs}$ is less than the $RMS_{ObsAnom}$ for the nominal depth bins.

The $\text{RMSD}_{Analysis-Obs}$ in potential density is reduced to about half of the $\text{RMSD}_{Freerun-Obs}$ in the upper 500m; the upper layer that is most effected by mesoscale eddies. The $\text{RMSD}_{Analysis-Obs}$ in potential density peaks in the upper 100m at 0.23 kgm$^{-3}$ and decreases gradually to below 0.1 kgm$^{-3}$ at 500m depth, remaining below that for the Argo-observed ocean deeper than 500m.

### 4.3.5   Velocities from moorings

Profiles of the complex correlation between the velocities from the free-running model and the analyses at the mooring velocity measurement locations are shown in Figure 13. The correlations are generally considerably improved in the analyses. The complex correlations for the analysis velocities approach the value of one for the South East Queensland shelf moorings (SEQ200 and SEQ400), which are on the shelf and shelf slope at the latitude where the EAC is found to be most coherent. At this same latitude, the deep water array moorings 1 to 4 (EAC1-4) have high complex correlations between the analysis and the

observations in the upper 400m of the water column. This is where the mean EAC jet is the strongest (refer to Figure 3). The EAC5 mooring is outside of the main jet and influenced by a more variable eddy-dominated circulation and the analysis has slightly lower correlations in the upper water column at this location. Further south on the shelf, velocity estimates are improved with depth-averaged free run and analysis complex correlations of 0.68 and 0.91, respectively, for the Coffs Harbour mooring (CH100), 0.37 and 0.84 for the Sydney mooring (SYD100) and 0.36 and 0.87 for the other Sydney mooring (SYD140).

### 4.3.6   Surface Velocities from HF radar

Here we choose to present the results in terms of surface velocities (rather than the scalar radial current speeds) as they are more meaningful in terms of the ocean surface currents. The observed surface velocities are computed from the assimilated radials and the corresponding values computed from the radial values extracted from the free-running model and the analyses. The complex correlations between these observed surface velocities and the surface velocities computed from the free-running model and the analysis are shown in Figure 14. Note that the complex correlation for a particular grid cell requires velocities to be computed, which requires radial data from each of the two sites to be available in that cell and that the beams overlap with an angle greater than $30°$ (velocity calculations where beam intersection angles are smaller than this are deemed inaccurate). Radial data is assimilated at other times but cannot be converted to velocities. Only grid cells where velocity values can be computed at more that 10 times over the 2-year period are included in the plots. In the free run, velocity estimates were best on the shelf and shelf slope with complex correlations reducing offshore of the shelf slope. Velocity is very well represented in the analysis under the HF radar footprint, with complex correlations from 0.8-1 across the entire footprint.

In terms of the radial current speeds measured from both NNB and RRK sites, the $\text{RMSD}_{Freerun-Obs}$ is 0.1-0.4 ms$^{-1}$ inshore of the 200m depth contour, 0.2-0.6 ms$^{-1}$ above the shelf slope (between the 200-2000m depth contours) and 0.3-0.5 ms$^{-1}$ offshore of the 2000m depth contour. The $\text{RMSD}_{Analysis-Obs}$, for both NNB and RRK sites, is between 0.1-0.25 ms$^{-1}$ across the entire radar footprints. The ratio of $\text{RMSD}_{Freerun-Obs}/RMS_{ObsAnom}$ is 0.5-1, reduced to 0.2-0.5 for the ratio of the $\text{RMSD}_{Analysis-Obs}/RMS_{ObsAnom}$.

### 4.4   Reanalysis Comparison to Independent Observations

Because IS4D-Var uses the model dynamics to solve for the increment adjustments, information from observed variables can propagate to unobserved regions such that the ocean state better fits and is in balance with the observations. Comparison of the reanalysis with independent, non-assimilated, observations allows us to assess the performance of the state estimate away from assimilated observations. As the principal aim of this work was to assimilate the maximum number of available observations in the region in order to provide a 'best estimate' of the ocean state over the 2-year period, few independent observations remain available for this comparison.

The available independent observations are from shipboard CTD casts that were taken on three separate cruises within the model domain over the 2-year period. 15 CTD casts were taken as part of the deployment of the EAC array, along the EAC array transect from 21-27th April 2012 (blue diamonds in Figure 15b). 5 casts were taken off of Sydney between 34.3-36.4° S and 151.6-152.8° E from 27-28 Feb 2013 (magenta diamonds in Figure 15b). 28 CTD casts were taken in two transects off

of Brisbane at 26.3° S and 27.1° S out to 155.8° E between 21-31 Aug 2013 (green diamonds in Figure 15b). The CTD cast observations are mapped to the model vertical levels for consistent comparison given the vertical discretisation of the model, and the corresponding model values extracted from the 2yr free run and the analysis. The $RMS_{ObsAnom}$, $RMSD_{Freerun-Obs}$ and $RMSD_{Analysis-Obs}$ for potential density in nominal depth bins for all CTD casts are shown in Figure 15a. In the upper 350m of the water column, the $RMSD_{Analysis-Obs}$ in the potential density is reduced to about half of $RMSD_{Freerun-Obs}$. For all CTD casts in the upper 200m, where the number of observations is the greatest (not shown), the depth-averaged $RMSD_{Freerun-Obs}$ of 0.33 kgm$^{-3}$ is reduced to 0.17 kgm$^{-3}$ in the analysis. This shows that a marked improvement in the representation of the subsurface ocean, as observed by these CTD casts, is achieved in the reanalysis.

Note that the profiles of Argo $RMSD_{Freerun-Obs}$ and $RMSD_{Analysis-Obs}$ in potential density (Figure 12) are similar to the RMSD profiles for the independent shipboard CTD observations. The reanalysis showing similar residual reduction for the assimilated Argo observations and for the non-assimilated CTD cast observations suggests a well-specified assimilation system in which a dynamically-balanced ocean state estimate is achieved with improved state-estimation throughout the model domain, rather than over-fitting to assimilated observations.

## 5  Summary and Conclusions

We have presented the development of a data assimilating model of the EAC region and assessed the performance of the corresponding reanalysis over a 2-year period. We use an advanced variational data assimilation scheme to integrate a state-of-the-art coastal ocean model with an unprecedented observational data for the southeast Australian region. We show that the free-running numerical model reproduces the long-term mean surface and subsurface ocean properties and represents the eddying circulation as expressed by the sea surface variability well. For the reanalysis, we show that the SSH and SST have mean RMS residuals with the observations of 7.6cm and 0.4° C. The RMS residual profile for temperature has a subsurface maximum of 0.9° C for Argo float observations, 0.9° C for ocean glider observations and 0.8° C for XBT observations. Surface and subsurface velocity observations from HF radar, shelf and offshore moorings match well with complex correlations between 0.8-1 in the upper 500m. The reanalysis has an RMS residual in potential density with independent (non-assimilated) shipboard CTD cast observations of under 0.2 kgm$^{-3}$ throughout the water column.

The performance of the reanalysis is dependent on prior assumptions of the model background and observation error co-variances. We processed the observations to be assimilated to eliminate fine-scale processes not resolved by the model, and carefully specified the prior observation and model background uncertainties. Overall, the prior uncertainty assumptions are considered reasonable and the assimilation achieves reduced analysis uncertainty by reduction of the cost function for each assimilation interval

Not only does the reanalysis provide a good fit to observations, it is the first reanalysis of the EAC region that resolves the continental shelf along southeast Australia (BRAN3 has a resolution of 10km (Oke et al., 2013), the shelf is 15km wide at its narrowest point) and the first high-resolution reanalysis of the region that uses the model physics to adjust the model in a dynamically consistent way (Zavala-Garay et al. (2012) has a resolution of 18-30km). Furthermore, it is the first attempt to

assimilate such a wide variety of observations in the region, including observations from moorings on and off the continental shelf, a coastal HF radar array and ocean gliders. The high-resolution and dynamic-consistency of the reanalysis mean that it has the potential to provide a marked improvement in our ability to capture important circulation dynamics in the EAC.

The reanalysis is being used to study the 3-dimensional structure of the current and the processes that drive its separation from the coast and eddy formation. Several modelling studies of coastal regions in south-eastern Australia are making use of the reanalysis for boundary forcing. Output from the adjoint model integrations performed in each assimilation interval is being used to directly assess the impact of specific observations on the estimates of circulation dynamics of interest. Through this we hope to understand which observations are most effective at improving our state-estimates and which locations are most effective to observe, providing valuable information on how we might improve the observing system to ultimately improve prediction.

## 6   Data availability

Model initial conditions and boundary forcing comes from the Bluelink ReANalysis version 3p5 (BRAN3, Oke et al. (2013)). Surface forcing is provided by the Australia Bureau of Meteorology's Australian Community Climate and Earth-System Simulation (ACCESS) 12km product (Puri et al., 2013). The observations used for assimilation are available through the Australian Integrated Marine Observing System's (IMOS) data portal (www.imos.org.au).

The reanalysis output is saved as snapshots of 3-dimensional fields of ocean properties (sea-level, temperature, salinity, velocities) every 4 hours over the 2-year period (2012-2013). The data is archived at UNSW Australia and can be made available for research purposes (contact the corresponding author of this paper).

*Acknowledgements.*   This research and Dr. C. Kerry were supported by an Australian Research Council Discovery Project #140102337. Data was primarily sourced from the Integrated Marine Observing System (IMOS) – IMOS is a national collaborative research infrastructure, supported by the Australian Government. We thank Dr. Holly Sims from the BOM for making ACCESS available to us and Dr. Gary Brassington, also from the BOM, for providing data in the preliminary phases of this work and for his useful discussions. The XBT data was kindly provided by Ken Ridgway of CSIRO Hobart.

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

**Table 1.** Total full-water-column alongshore transport (Sv) through 27.5° S (EAC deep water array), 30.3° S (Coffs Harbour) and 33.9° (Sydney) cross-shore sections as shown in Figure 3, computed daily for the 10-year free-running model period. The transport is computed for distances offshore of 266km, 296km and 276km, respectively, to span the location of the main jet.

| | | Transport (Sv) |
|---|---|---|
| **EAC Array (27.5° S)** | mean | -14.3 |
| | std | 28.4 |
| | min | -97.9 |
| | max | 59.5 |
| **Coffs Harbour (30.3° S)** | mean | -21.9 |
| | std | 31.7 |
| | min | -120.2 |
| | max | 66.5 |
| **Sydney (33.9° S)** | mean | -6.9 |
| | std | 39.2 |
| | min | -163.0 |
| | max | 117.8 |

Weaver, A., Vialard, J., and Anderson, D.: Three- and four-dimensional variational assimilation with a general circulation model of the tropical Pacific Ocean. Part I: Formulation, internal diagnostics, and consistency checks., Mon. Weather Rev., 131, 1360–1378, 2003.

Whiteway, T.: Australian Bathymetry and Topography Grid. Scale 1:5000000., Geoscience Australia, Canberra., 2009.

Wilkin, J. L., Bowen, M. M., and Emery, W. J.: Mapping mesoscale currents by optimal interpolation of satellite radiometer and altimeter data, Ocean Dynam., 52, 95:103, 2002.

Wyatt, L. R.: Error analysis for ACORN HF radars, Integrated Marine Observing System - Australian Coastal Ocean Radar Network, ACORN report 2014-2, 2014.

Zavala-Garay, J., Wilkin, J. L., and Arango, H. G.: Predictability of mesoscale variability in the East Australian Current given strong-constraint data assimilation, Journal of Physical Oceanography, 42, 1402–1420, 2012.

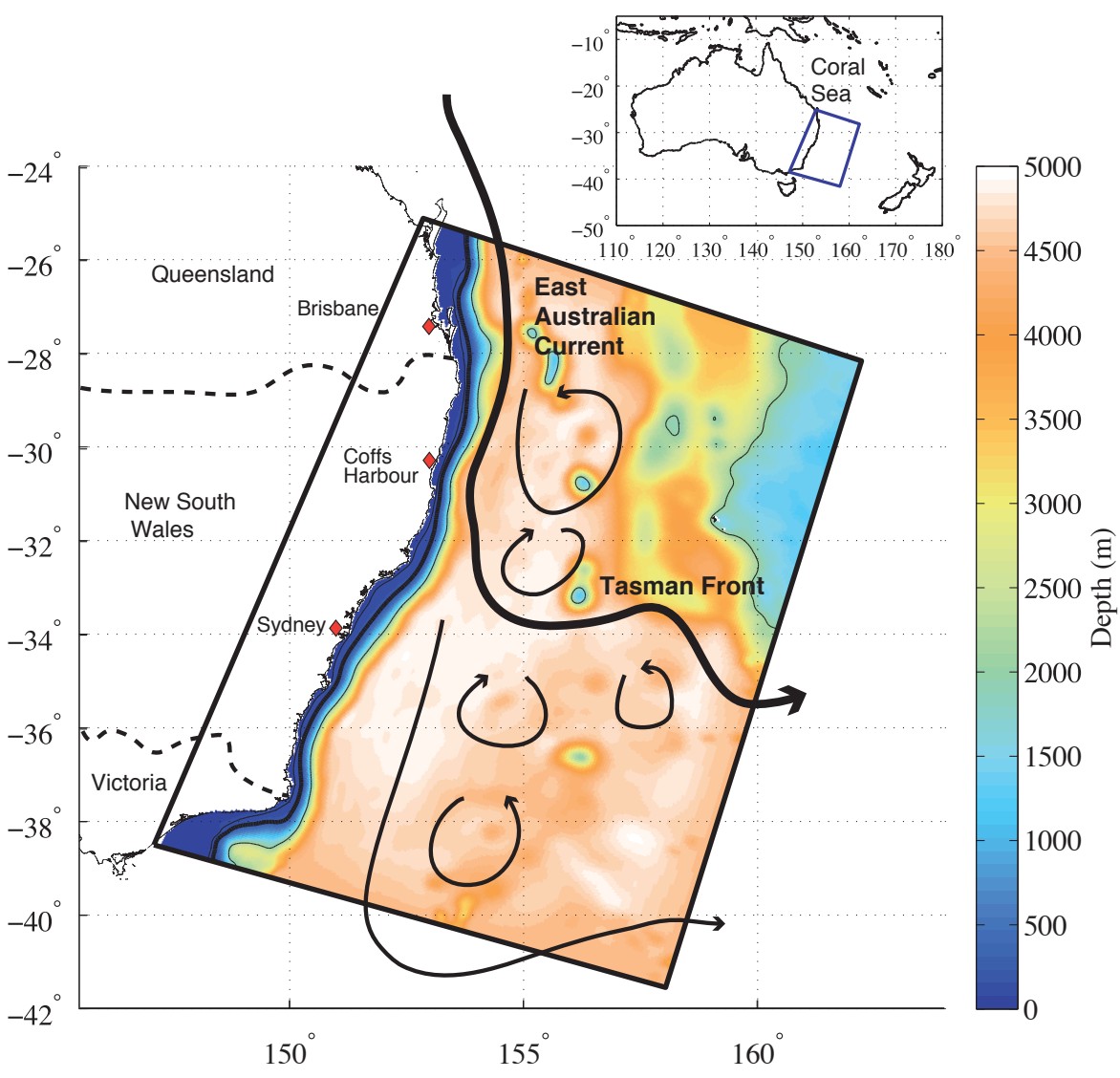

**Figure 1.** Model domain and bathymetry with the 100m, 200m (bold) and 2000m contours. Australian States are labelled and main towns are labelled and shown by the red diamonds. A cartoon of the EAC is overlain showing the typical separation latitude and the Tasman Front.

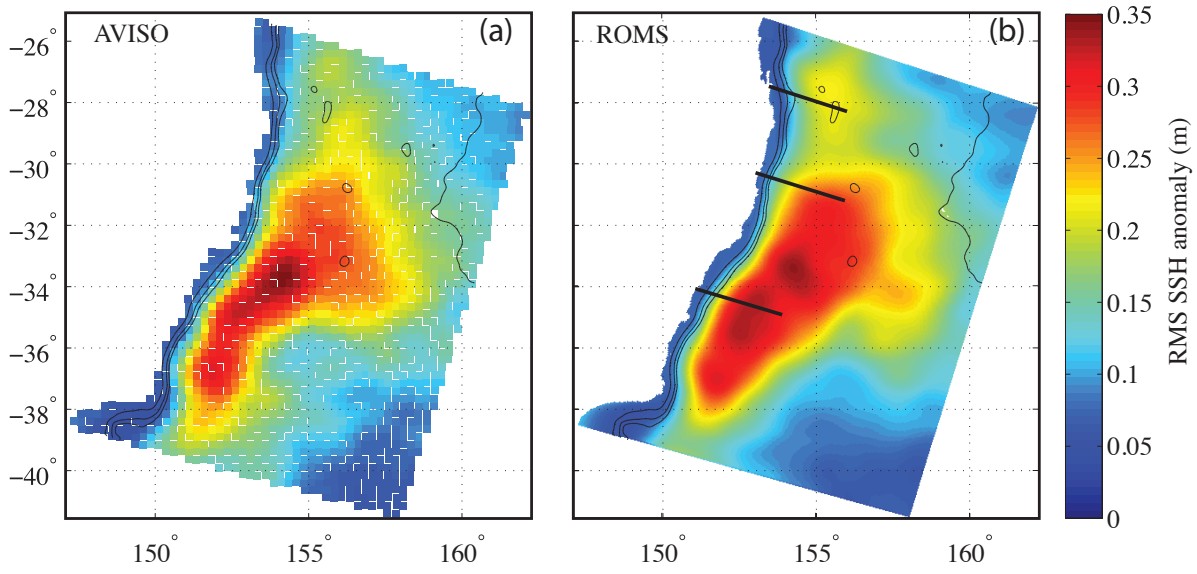

**Figure 2.** Root mean squared (RMS) SSH anomaly over 10 year period from AVISO, (a), and ROMS 10yr free run, (b). Cross sections plotted in Figure 3 and used for transport calculations in Table 1 are plotted on panel (b).

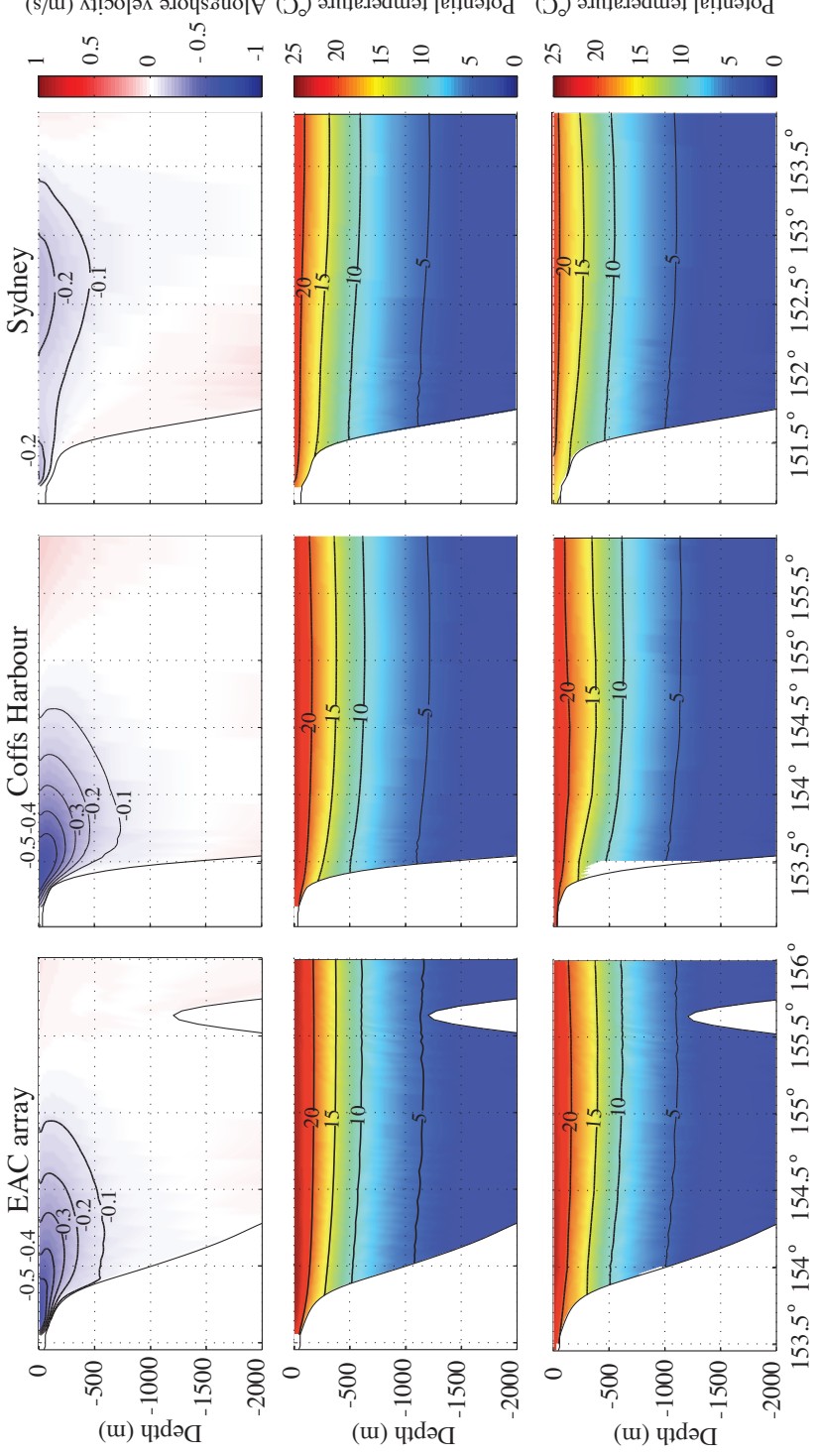

**Figure 3.** Mean alongshore velocity from the ROMS 10yr free run at the cross-shore sections that cross the coast at the EAC Transport array (27.5° S), Coffs (30.3° S) and Sydney (33.9° S), top row. Mean temperature from the ROMS 10yr free run, middle row, and mean temperature from CARS climatology, bottom row, for the same cross-shore sections.

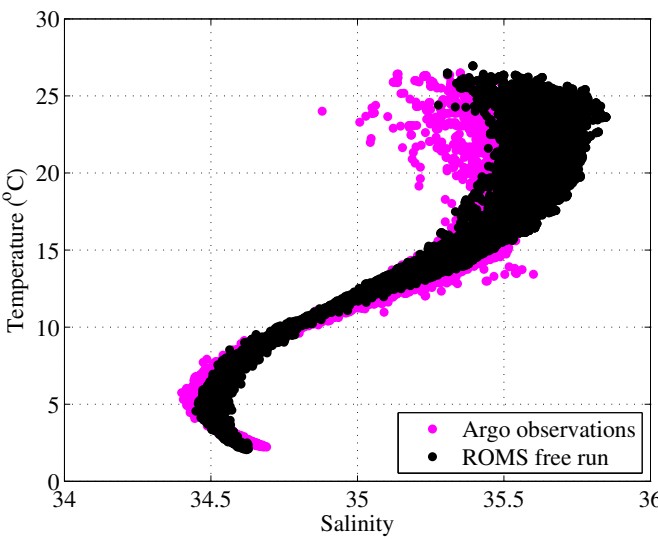

**Figure 4.** Temperature-Salinity diagram for the Argo observations and corresponding values from the 2yr free run for 2012-2013.

**Table 2.** Mooring information for the EAC deep water array moorings (EAC1-5), South East Queensland shelf moorings (SEQ200, SEQ400) and the NSW shelf moorings (CH100, SYD100, SYD140).

| Name | Lat (° S) | Lon (° E) | Water depth (m) | Distance offshore (km) | Temporal coverage | Sensor Depth Range (m) | | |
|---|---|---|---|---|---|---|---|---|
| | | | | | | *Temperature* | *Salinity* | *Velocity* |
| **CH100** | 30.27 | 153.40 | 98 | 25 | 1 Jan 2012 - 30 Dec 2013 | 5-100 | - | 9-89 |
| **SYD100** | 33.94 | 151.38 | 104 | 10 | 1 Jan 2012 - 30 Dec 2013 | 11-107 | - | 1-99 |
| **SYD140** | 33.99 | 151.45 | 138 | 19 | 1 Jan 2012 - 30 Dec 2013 | 21-143 | - | 24-129 |
| **EAC1** | 27.31 | 153.97 | 1525 | 53 | 21 Apr 2012 - 23 Aug 2013 | 60-1060 | 60-1060 | 43-1054 |
| **EAC2** | 27.31 | 153.99 | 1940 | 55 | 22 Apr 2012 - 24 Aug 2013 | 163-1045 | 163-1045 | 9-1495 |
| **EAC3** | 27.25 | 154.29 | 4220 | 85 | 23 Apr 2012 - 24 Aug 2013 | 156-3991 | 156-3991 | 9-3968 |
| **EAC4** | 27.21 | 154.65 | 4745 | 121 | 25 Apr 2012 - 25 Aug 2013 | 154-4009 | 154-4009 | 38-3974 |
| **EAC5** | 27.10 | 155.30 | 4797 | 185 | 26 Apr 2012 - 26 Aug 2013 | 192-1109 | 192-1109 | 107-4016 |
| **SEQ400** | 27.33 | 153.88 | 405 | 44 | 01 Apr 2012 - 06 Jun 2013 | 48-375 | 48-375 | 23-405 |
| **SEQ200** | 27.34 | 153.77 | 209 | 33 | 01 Apr 2012 - 06 Jun 2013 | 40-189 | 40-189 | 23-196 |

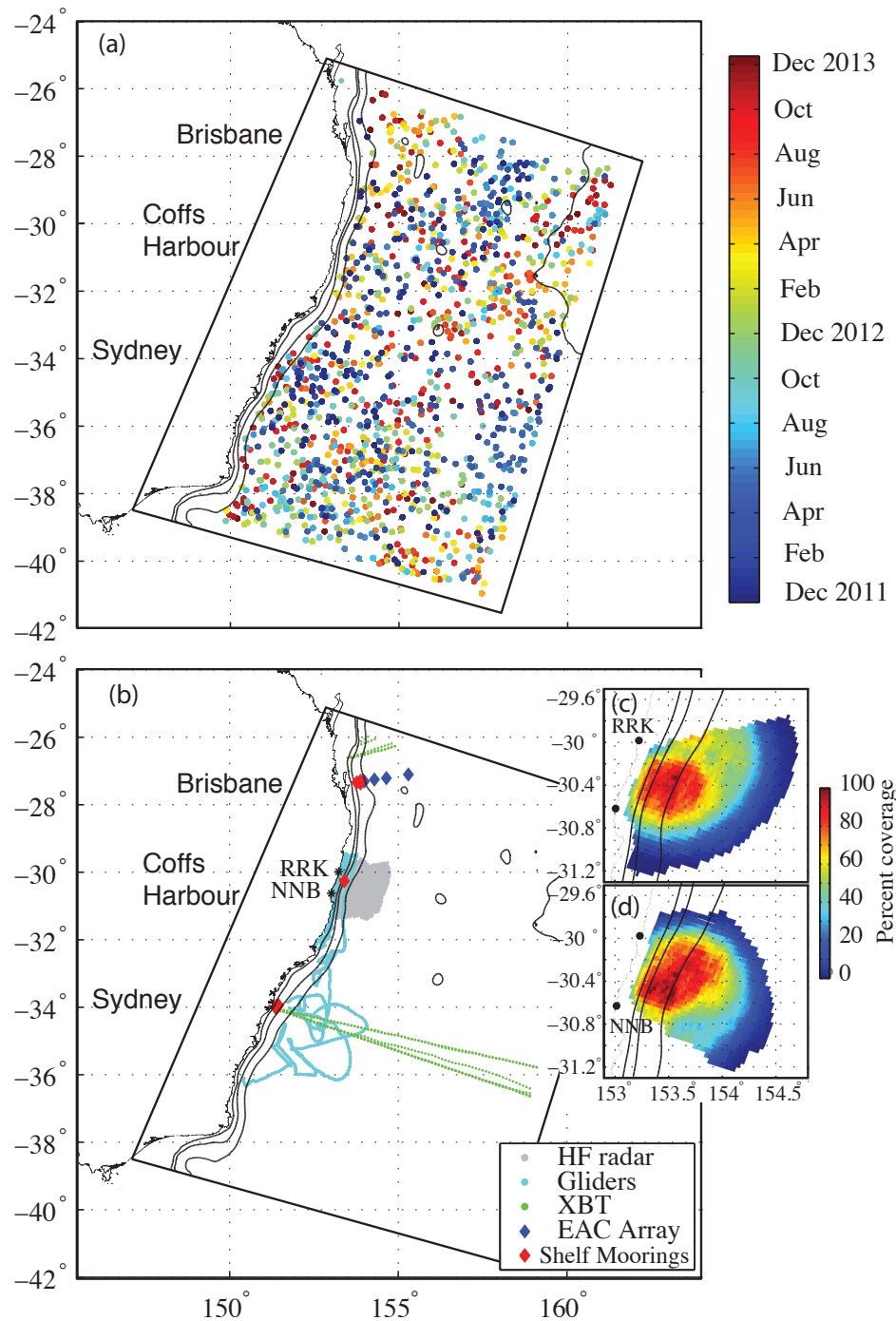

**Figure 5.** Argo observations coloured by time of occurrence, (a), and all other observations, with the exception of satellite-derived SSH and SST, (b). 100m, 200m and 2000m contours are shown. Coastal towns are labelled in line with their location on the coast. HF radar sites Red Rock (RRK) and North Nambucca (NNB) are shown with black asterisks in (b) and zooms showing the percent coverage of radial data for the two stations are shown in (c) and (d).

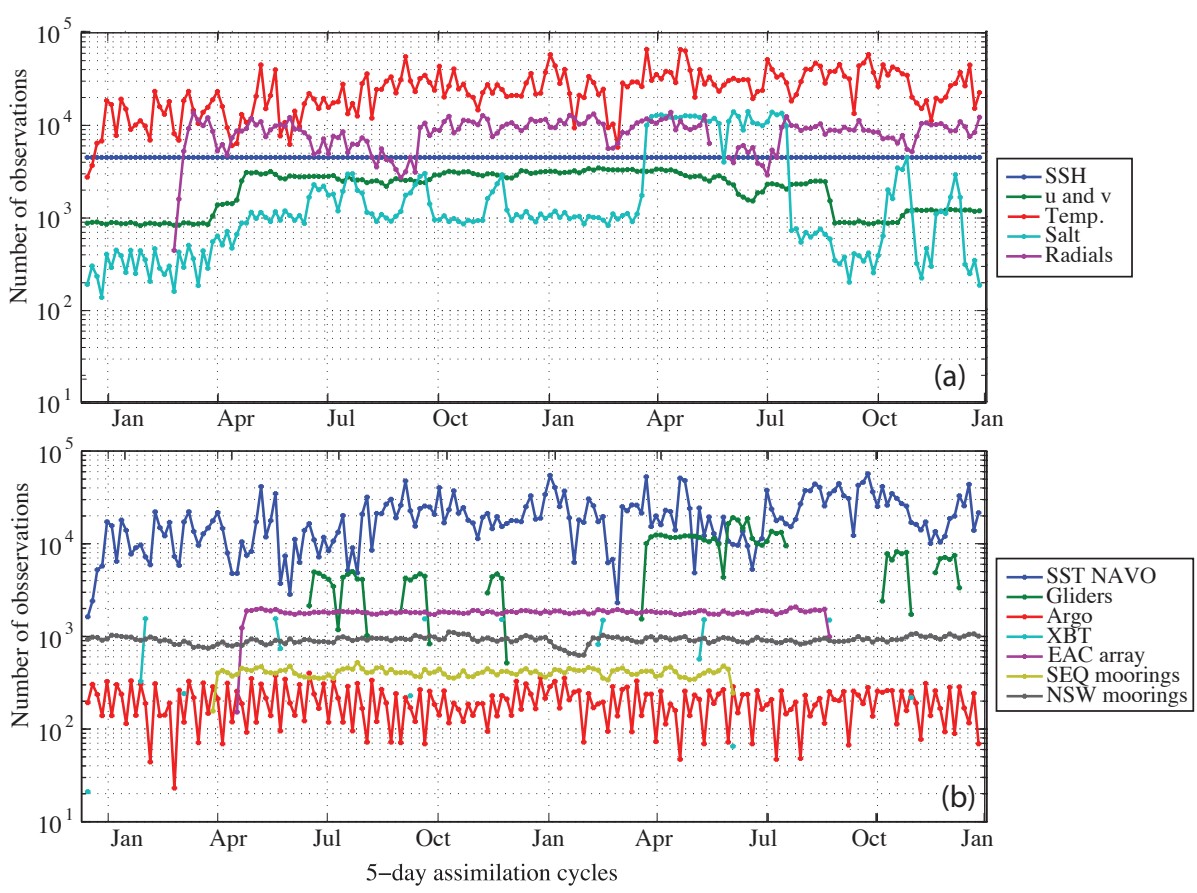

**Figure 6.** Number of observations (after processing) used in each 5-day assimilation window; for each observation type, (a), and temperature observations for each data source, (b).

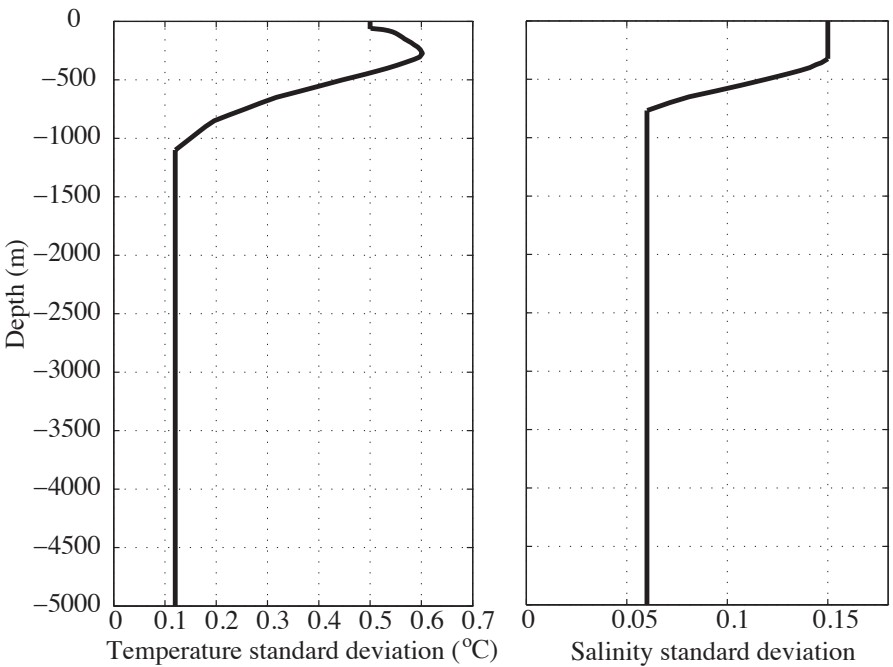

**Figure 7.** Nominal minimum observation uncertainty profiles applied to subsurface temperature and salinity observations offshore of the continental shelf.

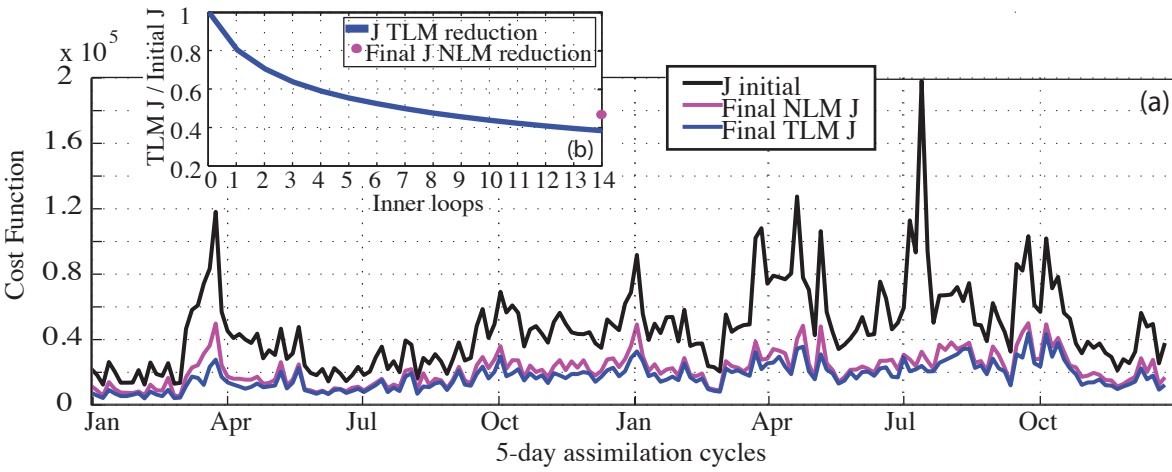

**Figure 8.** Initial nonlinear cost function and the reduction achieved in the final (14th) tangent linear model *inner loop* and the final nonlinear cost function, plotted for each assimilation interval, (a). Mean cost function reduction for each of the 14 *inner loops* for all 5-day assimilation intervals, (b).

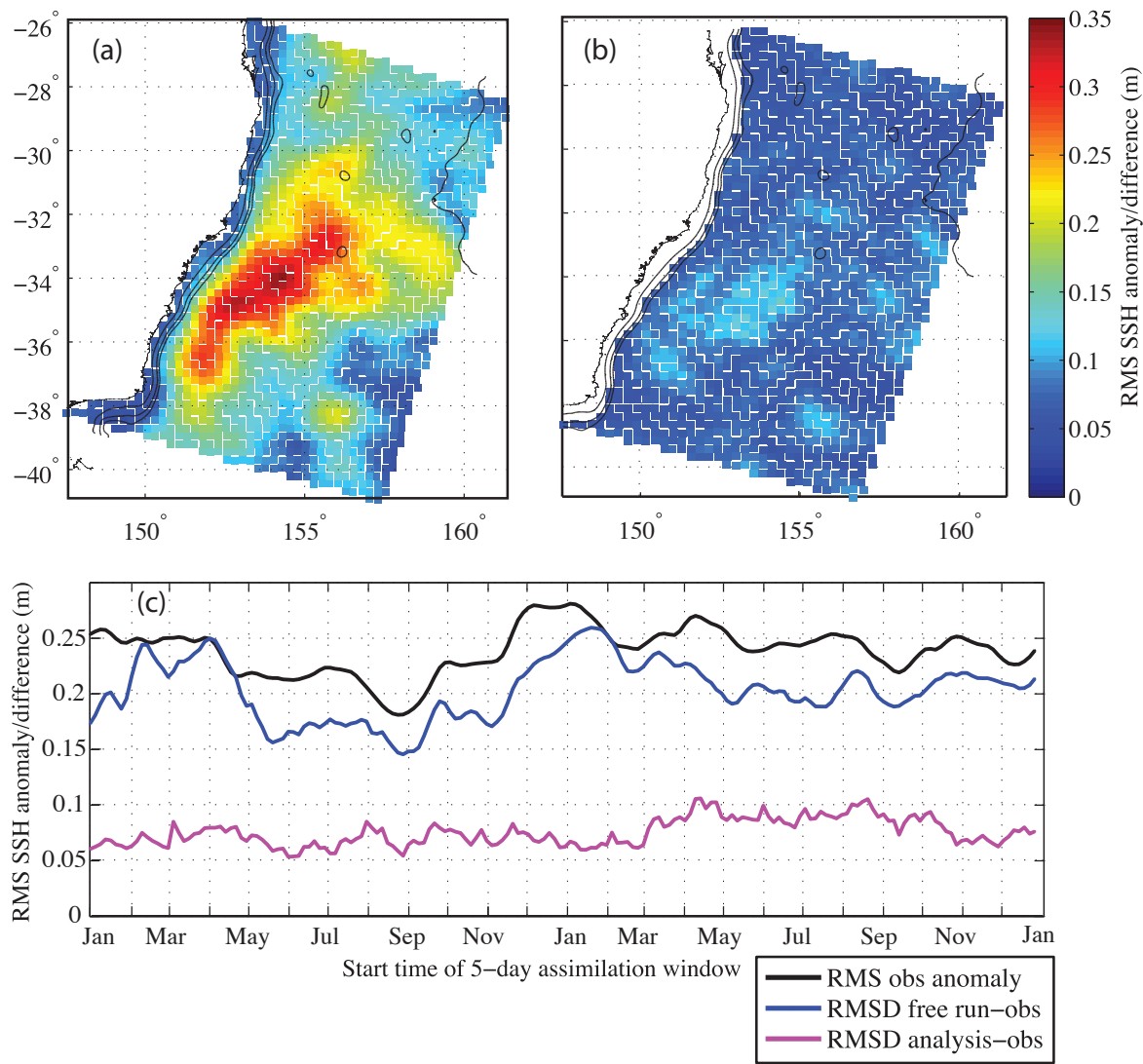

**Figure 9.** RMS SSH observation anomaly (a) and RMS SSH difference between the analysis and observations (b) for the 2-year assimilation window. Time-series of spatially-averaged RMS SSH observation anomaly, RMS SSH difference between the free run and observations, and RMS SSH difference between the analysis and observations, for each assimilation window (c).

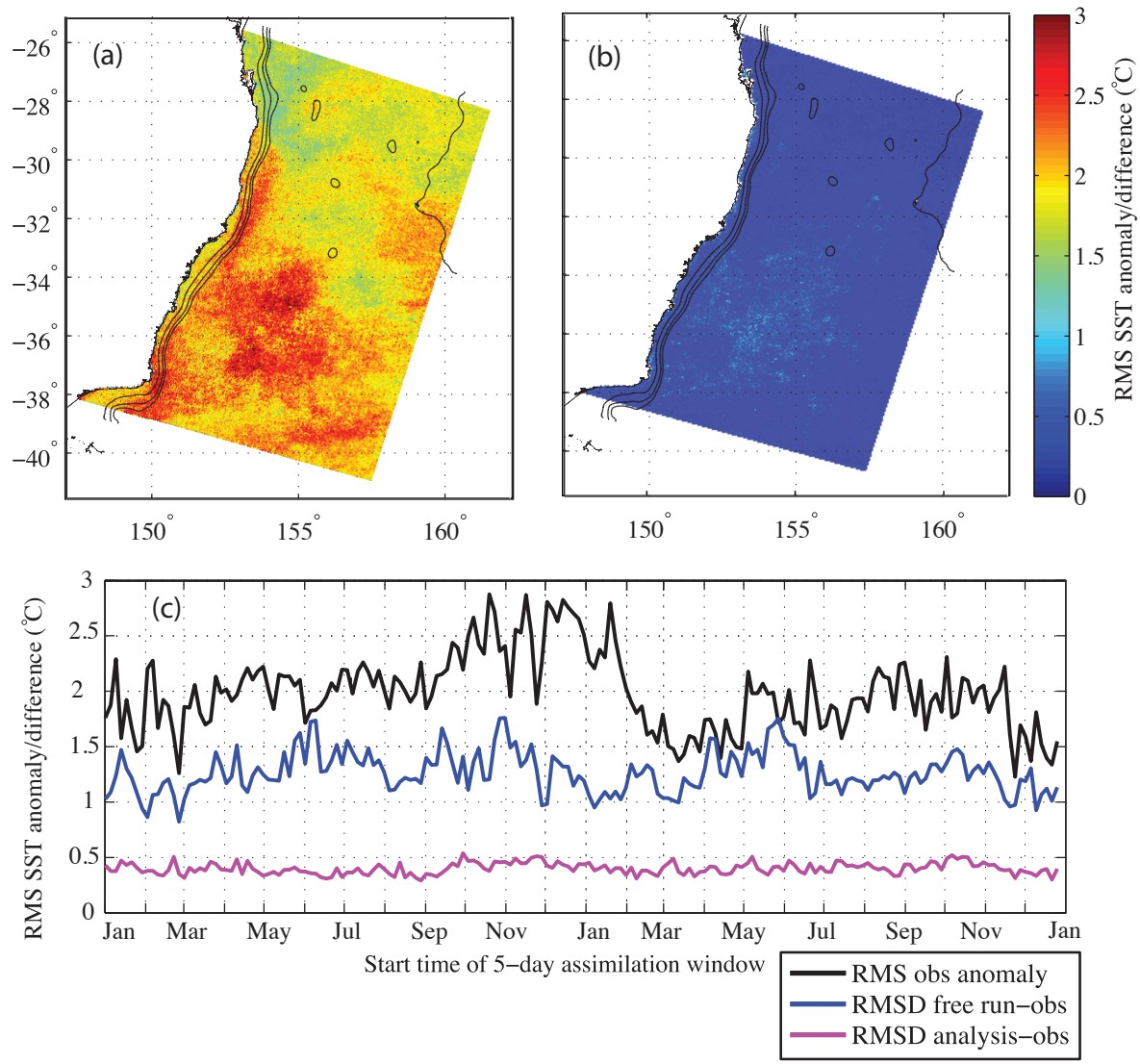

**Figure 10.** RMS SST observation anomaly, including seasonal cycle, (a) and RMS SST difference between the analysis and observations (b) for the 2-year assimilation window. Time-series of spatially-averaged RMS SST observation anomaly, RMS SSH difference between the free run and observations, and RMS SSH difference between the analysis and observations, for each assimilation window (c).

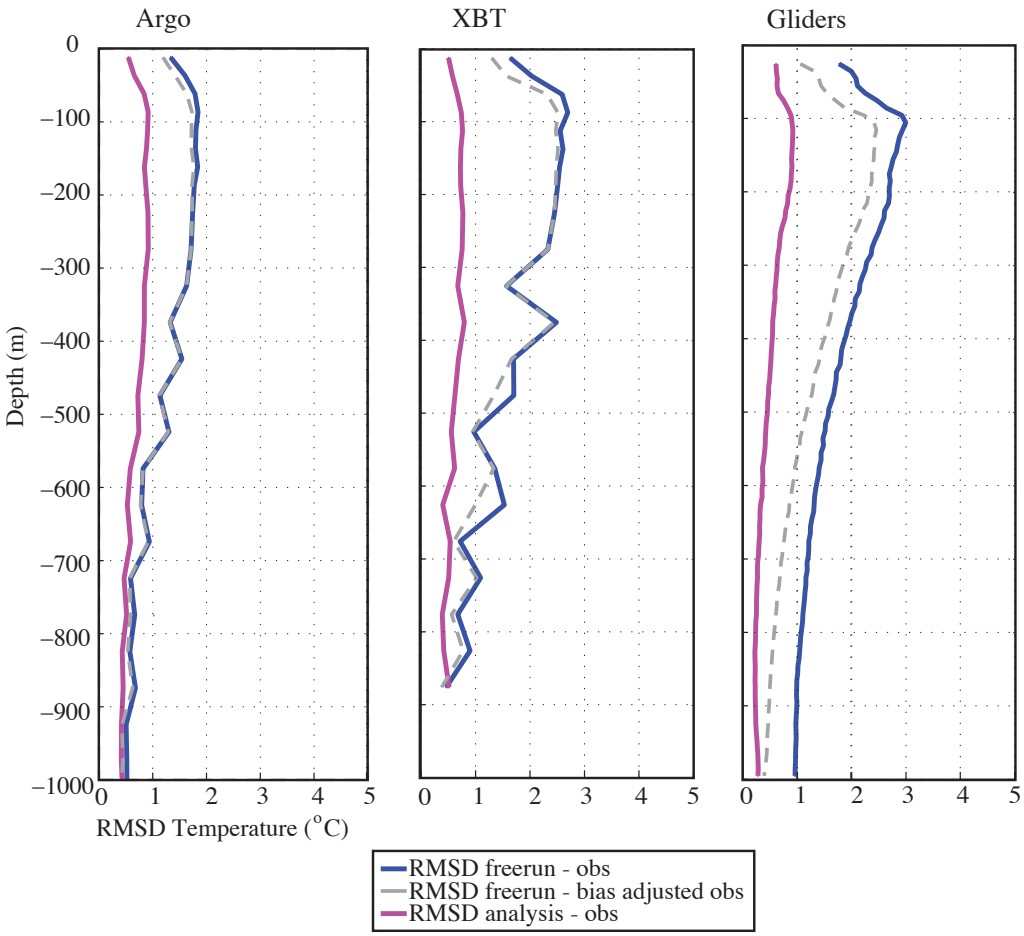

**Figure 11.** RMS difference between the free run and observations, the free run and the bias adjusted observations, and the analysis and observations for Argo (a), XBT (b) and Glider (c) observations in nominal depth bins for the 2-year assimilation window. Argo and XBT depth bins are 25m from the surface to 200m and 50m below 200m, Glider bins are 10m throughout the water column.

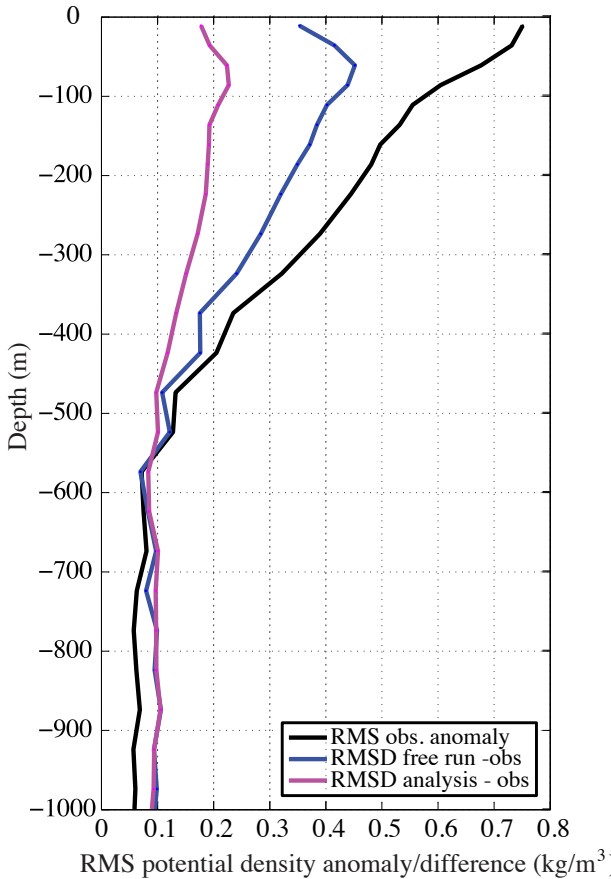

**Figure 12.** RMS potential density observation anomaly and RMS difference between the free run and observations, and the analysis and observations for Argo float observations. Observations are grouped into nominal depth bins of 25m from the surface to 200m and 50m below 200m.

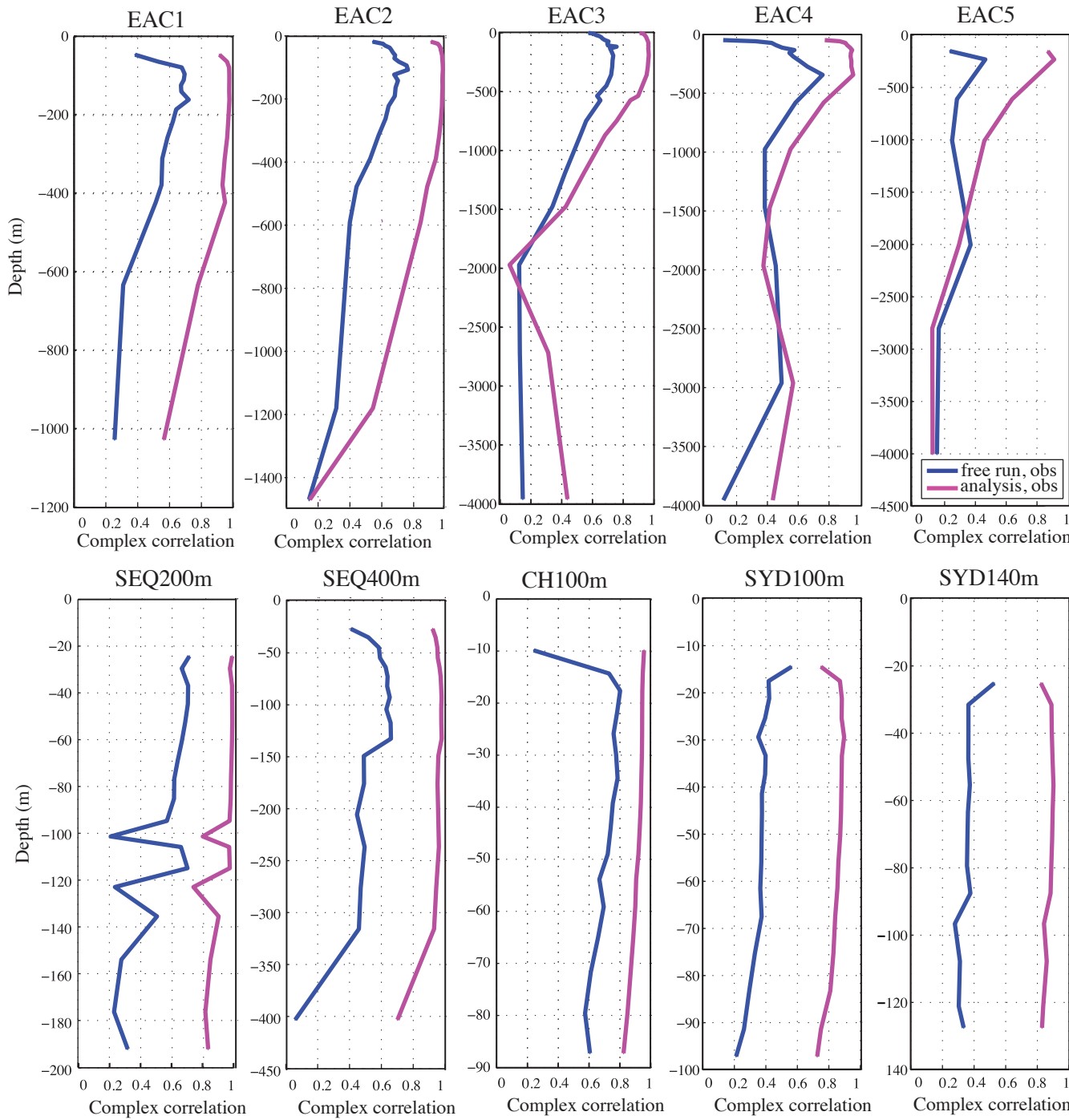

**Figure 13.** Complex correlation between observed velocities and free run and analysis velocities at mooring locations.

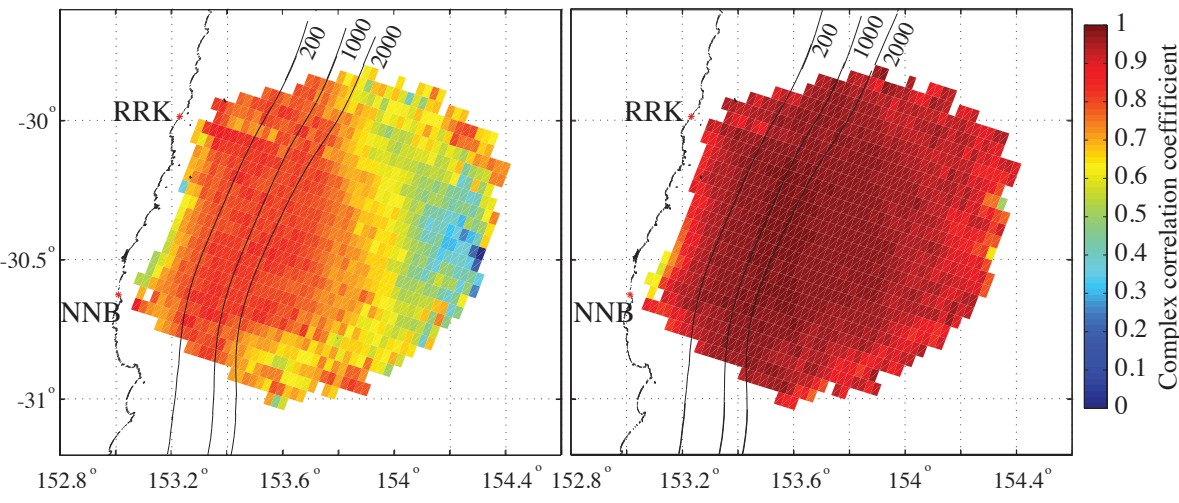

**Figure 14.** Complex correlation of surface velocities computed from the assimilated HF radar radials, and surface velocities computed from the corresponding free run (a) and analysis (b) radials. 200m, 1000m and 2000m bathymetry contours are shown.

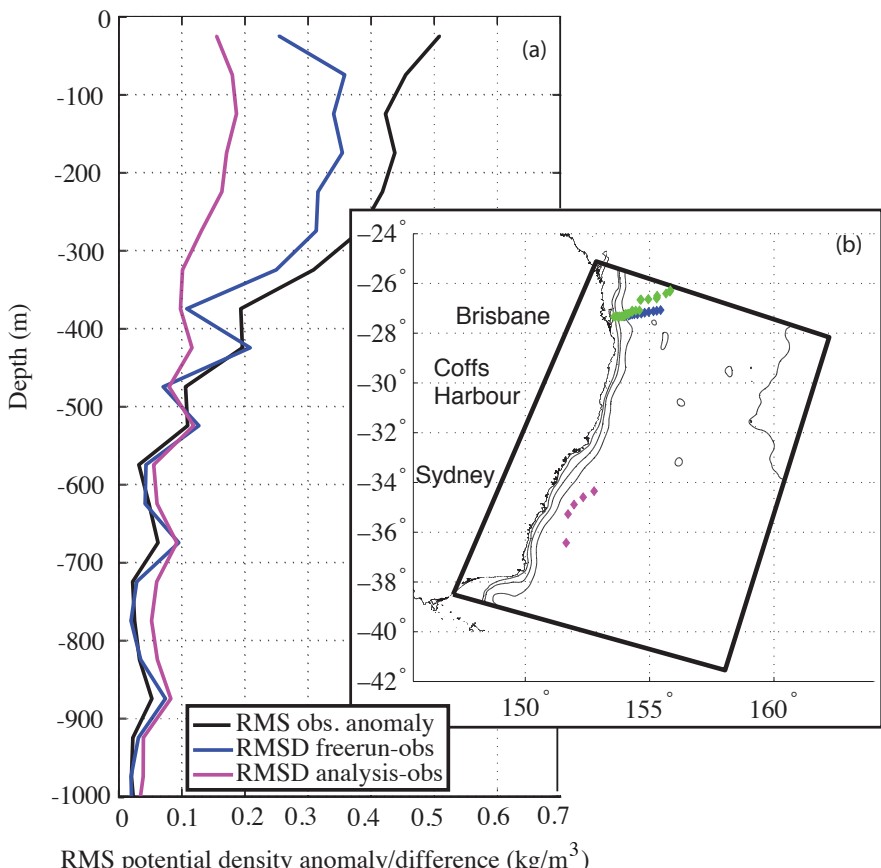

**Figure 15.** RMS potential density observation anomaly and RMS difference between the free run and observations, and the analysis and observations for independent CTD cast observations mapped to model vertical levels, (a). Observations are grouped into nominal depth bins of 50m. Locations of the CTD casts for the three separate cruises, described in Section 4.4, are shown in (b).