# Peer review of "Development and evaluation of a high-resolution reanalysis of the East Australian Current region using the Regional Ocean Modelling System (ROMS 3.4) and Incremental Strong-Constraint 4-Dimensional Variational data assimilation (IS4D-Var)"

_Geoscientific Model Development, 2016_

## Referee Comment (RC1) · Anonymous Referee #1 · 4 Apr 2016

General Comments:

This paper presents the results of an impressive suite of calculations using the ROMS variational data assimilation system to compute a series of ocean circulation estimates for the East Australia Current. The observational coverage appears to unprecedented, and the authors have apparently done a fairly rigorous job of tuning the system and evaluating the system performance.

Overall, this is a well written paper, although it suffers in a few places from being imprecise (detailed comments below).

Specific comments:

(1) Sections 3.2 and 3.3 In general I found sections 3.2 and 3.3 to be unsatisfying. To the 4dvar expert they are not really very illuminating, and to the non-expert they conveys no real useful information in that technical terms like "inner loops", "outer loops" and "cost function" are used with no useful context. I recommend that the authors rework these sections, perhaps adding an equation or two - this would help to clarify the text. More technical details could also go in an appendix. In contrast the authors devote a great deal of text and detail to the observations, but very little to the 4dvar machinery which is doing all the heavy lifting here.

(2) Section 4.1 This section discusses the consistency between the a priori specified error variances for the background and observations, and those diagnosed a posteriori from the innovation statistics following the methods introduced by Descroziers et al (2005). There is a problem with the language in this section that needs to cleared up. Throughout, the authors refer to "posterior errors" when what they really mean are the "diagnosed prior errors." The term "posterior errors" implies that these are the errors in the analysis, but that is not what is being computed here.

Detailed and technical comments:

Page 1, line 1: "inherently dynamic" - are all circulations inherently dynamic, not just the EAC?

Page 1, line 19: "model dynamics to determine covariance" - what do you mean by the statement?

Page 2, line 14: Reword to say "submesoscale and mesoscale eddies"

Page 2, line 16: You say that barotropic and baroclinic instabilities are unpredictable - this is not true in either the atmosphere or the ocean.

Page 2, line 26: "adjoint to compute covariance" - what do you mean here (and elsewhere)?

[Figure]

Page 2, line 29: "without requiring ensemble or long-run statistics" - this seems like a strange statement in this sentence.

Page 3, line 14: Why is High Frequency capitalized here?

Page 3, line 32: "eddying general circulation" - clumsy wording.

Page 5, line 6: "clamped in the baroclinic" - this does not make grammatical sense.

Page 5, lines 9 and 10: Why does the heat flux need to be consistent with BRAN3? BRAN3 is only being used at the open boundaries right?

Figure 3: Instead of plotting both the ROMS solution and CARS, why not plot ROMS and (ROMS-CARS). From the difference plots it will be clearer where the model is deficient and where it is doing well.

Page 6, lines 22-24: Does the 2 year free run start from the end of the 10 year free run?

Page 7, line 10: Reword as "... deviations OF THE MODEL from the observations..."

Page 7, line 11: Reword as "J comprises a term that represents the difference..."

Page 7, line 14: Reword as "covariance, and a term that penalizes...."

Page 7, line 34: "free within the known uncertainties in the system" - this does not make grammatical sense.

Page 8, line 8: Reword as "...INDICATED that for this MODEL configuration, the linear ..."

Page 8, line 10: Replace "feasible" with "reasonable" or "affordable"

Page 9: lines 13, 15 and elsewhere: You imply here, and elsewhere, that the model is capturing submesoscale variability. The submesoscale is generally viewed as being in the 1-10km scale range, so your model will not resolving the submesoscale. You should remove the references to the submesoscale circulation.

Page 9, line 23: Reword as "... observation ERROR VARIANCE for the assimilation is CHOSEN to BE the square..."

Page 9, line 29: "not resolved by the model" - I think you mean not resolved by the data

Page 11, line 23: "covariance" misspelled.

Page 13, line 17: lower than the model AND OBSERVATION prior

Page 14, line 19: Reword as "These climatological varinces provide..."

Page 14, line 22: You say here that because you have only estimates of the variances, the background covariance of each field is estimated as a diffusion operator. This is not the reason that covariances are estimated this way. I suggest you go back to the original papers on modeling covariances using diffusion equations and brush up on some of the ideas (e.g. balanced versus unbalanced flows, etc).

Page 14, lines 24 and 25: Moore et al (2011) is not appropriate reference - you should refer to the appropriate equations in the orginal paper by Desroziers et al (2005).

Page 15, line 4: The authors claim that they have generated a "near-optimum" minimization. This is most certainly not the case since the background and observation error covariances they use are very far from being the true error covariances. The authors should tone this down or remove it.

Page 15, line 13: Reword as $t_1, t_2, ... t_n$ instead as $t_1-t_2$ since this looks like the time difference.

Figure 12 and 13: You show only the complex correlations - what are the rms errors in the current speeds? It would interesting to know this also.

Page 18, line 18: Say "diagnosed" instead of "posterior" - see specific comment (2) above.

Figure 4: The light grey points are almost illegible (they are in legend).
All figure: The fonts are tooo small to legible in many cases (eg Fig. 3, Fig. 8, Fig. 9, Fig. 12). The legends in particular are difficult to read.

---

## Referee Comment (RC2) · Anonymous Referee #2 · 14 Apr 2016

Specific Comments: Page 2, Line 18: Sentence beginning with: "Much of the effort . . ." This sentence is awkward to read. But more importantly see: O'Kane, T.J., Oke, P.R. and Sandery, P.A., 2011. Predicting the east Australian current. Ocean Modelling, 38(3), pp.251-266. for additional work in this area.

Page 2, Line 25: You state that a major advantage of the 4D-Var system is the fact that it relies on a linearised version of the governing equations. It could be contested that this is in face a disadvantage in highly non-linear systems and the assimilation window needs to be shortened such the the system behaviour is quasi-linear. I suggest you

make it clear at this point that there are also some limitations (as all systems have), that need to be accounted for.

Page 5, Line 33: "downward tilt", this is not a very intuitive description of the thermocline position. e.g. Does this mean the thermocline is uplifted near the shelf break, or depressed near the shelf break?

Page 7, line 4: In this para you give a high level description of the way IS4D-Var work. I suggest you add an additional sentence explaining what the increment is. In the previous para you say the observations are used to constrain the model increments, again without introduce the concept of what an increment is. A sentence such as: "The IS4D-Var system generates a vector (time series) of increments that are added to the initial conditions (and forcing), to minimise the cost function."

Page 7, line 20: How many inner loops did you run, and what was time series for the "J ratio" or reduction in the cost function?

Page 8, line 16: How is the blended product constructed? Is this done my weighting in the assimilation scheme, or as a post processing step.

Page 8: There is some duplication of material between section 3.2 and section 3.3.

Page 8, line 29: A good discussion of representation error can be found in:

Oke, P.R. and Sakov, P., 2008. Representation error of oceanic observations for data assimilation. Journal of Atmospheric and Oceanic Technology, 25(6), pp.1004-1017.

Section 3.4.1: The AVISO product is a daily gridded product that is derived from along track altimetry. Why not use the raw along track altimetry and use the IS4D-Var system constrain the SSH? By using a daily gridded product, some cells will have a higher error due to the statistical interpolation, while others that lie on the satellite track, will have a low uncertainty, therefore, by assuming a 6cm error, you are degrading the along track product to the error estimate used for the interpolated cells. Why not use the IS4D-Var system to assimilate the along track data, then use the model dynamics

(which should be better than the statistical model) to constrain the unobserved cells? What is the implication of using the interpolated SSH cells with the observed temp and salinity (if/when this occurs), does J fail to converge?

Section 3.4.4: The errors that are assigned to Argo profiles appear very large. How can you justify the error on an Argo float being up to a factor of 3 times larger than an SST observation? Depending on the grid resolution of the model, that cannot just be written off as representation error. Do you need to artificially inflate the prior over the floats to allow J to converge due to the gridded AVISO product?

Page 13, Line 23: Is it usual in 4D-Var system to use a diffusion operator as a prior? Is this not the same as using an isotropic 3D Gaussian function? Is there a reference that can support your use of the diffusion operator?

Section 3.5 What is the implication of choosing sub-optimal de-correlation length scales for the reanalysis system.

Section 4.2 Are the observations used to calculate the RMSD'd in this section with-held from the assimilation system, or are you reporting the forecast error statistics (i.e. before assimilation takes place)? Is the RMSD reduction between the free-run and as-similating run due to a reduction in bias or due to the dynamics features being correctly predicted?

General Comments:

The second half of the introduction reads like a combination of the abstract, methods and results/conclusion section. To avoid replication of information, it would be worth-while considering moving the material from page 3, line 3 - line 25 into other more appropriate sections.

Can the the inner loops of the assimilation scheme generate unrealistic initial conditions or forcing data?

Is it great to see the Aquarius data being used, can you give any indication as to the

value of this data in the assimilation scheme? i.e. how much does it contribute to the reduction in J?

The authors should be commended in their use of such a wide variety of observational data to constrain the reanalysis. It would be fascinating to know the relative impacts of each of the observation platforms?

This study presents a detailed assessment of the reanalysis skill, but I was left wondering about the following points. These may be outside of the scope of this study, but I thought I'd pass them on as food for thought:

1.) The 10 year control run was assessed against the climatological EAC transports, and shown to give a good statistical representation of the mean transport. What was the difference in transport between the 2 year free run, and the reanalaysis. Is there a substantial difference between the estimated transports? Which observations had the most impact on quantifying the transport (e.g. see the study of Moore et al., 2011)? Such a study may provide insight into how an observing system should be configured to monitor and observe the impacts of Climate change on the EAC strength.

2.) What value does the high resolution reanalysis give compared to BRAN 3p5? Obviously, the high res reanalysis is needed for nesting purposes, but do the error statistics look different between the course global reanalysis and the high resolution regional reanalysis?

---

## Author Comment (AC1) · 30 May 2016

We thank the reviewers for their helpful comments and suggestions. We have taken these on board and feel that we have an improved manuscript that is ready for publication. Our responses to the specific comments are outlined below:

Referee #1:

*General Comments:*
*This paper presents the results of an impressive suite of calculations using the ROMS variational data assimilation system to compute a series of ocean circulation estimates for the East Australia Current. The observational coverage appears to unprecedented, and the authors have apparently done a fairly rigorous job of tuning the system and evaluating the system performance. Overall, this is a well written paper, although it suffers in a few places from being imprecise (detailed comments below).*

*Specific comments:*
*(1) Sections 3.2 and 3.3 In general I found sections 3.2 and 3.3 to be unsatisfying. To the 4dvar expert they are not really very illuminating, and to the non-expert they conveys no real useful information in that technical terms like "inner loops", "outer loops" and "cost function" are used with no useful context. I recommend that the authors rework these sections, perhaps adding an equation or two - this would help to clarify the text. More technical details could also go in an appendix. In contrast the authors devote a great deal of text and detail to the observations, but very little to the 4dvar machinery which is doing all the heavy lifting here.*

Section 3.2 has been greatly expanded to include more technical detail regarding the 4D-Var scheme, including several equations outlining more clearly the increment and the cost function, and explaining how the minimisation is achieved. For extensive detail on the 4D-Var scheme, which is outside of the scope of this paper, the reader is referred to Moore et al 2011c.

Section 3.2 is from page 7, line 12 to page 9, line 20 of the new revision.

Section 3.3 has also been expanded with reference to later sections of the manuscript where the configuration is described in detail and the consistency checked.

Section 3.3 is from page 9, line 21 to page 10, line 7 of the new revision.

*(2) Section 4.1 This section discusses the consistency between the a priori specified error variances for the background and observations, and those diagnosed a posteriori from the innovation statistics following the methods introduced by Descroziers et al (2005). There is a problem with the language in this section that needs to cleared up. Throughout, the authors refer to "posterior errors" when what they really mean are the "diagnosed prior errors." The term "posterior errors" implies that these are the errors in the analysis, but that is not what is being computed here.*

We thank the reviewer for this comment. In Section 4.1, the wording has been clarified to refer to the diagnosed errors. More detail explaining the diagnostics has also been

added to the first paragraph of Section 4.1. We have also included an additional diagnostic called the 'optimality' value, which we feel strengthens this section on the consistency of the prior uncertainty estimates.

Section 4.1 is from page 16, line 29 to page 17, line 25 of the new revision.

*Detailed and technical comments:*

*Page 1, line 1: "inherently dynamic" - are all circulations inherently dynamic, not just the EAC?*

We have replaced this phrase with 'highly variable'.
Page 1, line 1 of new revision.

*Page 1, line 19: "model dynamics to determine covariance" - what do you mean by the statement?*

We understand that this wording may not be intuitive and have therefore adjusted the text to read "as the assimilation system uses the model dynamics to adjust the model state estimate."
Page 1, line 21 of new revision.

*Page 2, line 14: Reword to say "submesoscale and mesoscale eddies"*

This has been reworded.
Page 2, line 14 of new revision.

*Page 2, line 16: You say that barotropic and baroclinic instabilities are unpredictable - this is not true in either the atmosphere or the ocean.*

We have reworded this to read with "Eddies are typically generated by barotropic or baroclinic instabilities which are difficult to forecast so ….."
Page 2. line 16 of new revision.

*Page 2, line 26: "adjoint to compute covariance" - what do you mean here (and elsewhere)?*

Again, we understand that this wording may not be intuitive to the reader. We have adjusted the text to read "In this work we use Incremental Strong-constraint 4-Dimensional Variational data assimilation (IS4D-Var), which generates increments to adjust the model initial conditions, boundary and surface forcings such that the difference between the model solution of the time-evolving flow and all available observations is minimised over an assimilation interval. The 4D-Var scheme uses the linearised model equations and their adjoint to compute the increment adjustments, such that the model is adjusted in a dynamically consistent way to minimise the difference between the observations and the modelled time-evolving ocean state."

Page 2, lines 25-29 of new revision.

*Page 2, line 29: "without requiring ensemble or long-run statistics" - this seems like a strange statement in this sentence.*

This has been deleted and the sentence now reads "Using the linearised equations allows dynamical connections between state variables to propagate information from observed variables to unobserved, dynamically-linked variables."
Page 2, line 30-31 of new revision.

*Page 3, line 14: Why is High Frequency capitalized here?*

This has been replaced with "high-frequency".
Page 3, line 17 of new revision.

*Page 3, line 32: "eddying general circulation" - clumsy wording.*

This has been replaced with "atmospherically-forced eddying ocean circulation".
Page 4, line 3 of new revision.

*Page 5, line 6: "clamped in the baroclinic" - this does not make grammatical sense.*

This has been replaced with "For the assimilation, the baroclinic boundary conditions at all three ocean boundaries are clamped to the BRAN3 boundary conditions".
Page 5, lines 11-12 in new revision.

*Page 5, lines 9 and 10: Why does the heat flux need to be consistent with BRAN3?*
*BRAN3 is only being used at the open boundaries right?*

This has been explained further in the text. "Because the BRAN3 system is run with different atmospheric forcing than we use, a correction was applied to the surface heat flux forcing such that the SST from BRAN3 is in balance with the atmospheric surface forcing for each month. This correction is applied so that the surface heat flux applied through the atmospheric forcing is in balance with BRAN3, which is providing the open boundary forcing."
Without this correction, the model may be receiving too much/too little surface heating compared to the heat received at the open boundaries. This would result in a temperature bias in the free-running model, where the surface waters became hotter/cooler than BRAN3 over the 2-year model run period.
Page 5, lines 16-19 of new revision.

*Figure 3: Instead of plotting both the ROMS solution and CARS, why not plot ROMS and (ROMS-CARS). From the difference plots it will be clearer where the model is deficient and where it is doing well.*

Figure 3:
The difference plots reveal some small differences, however these differences do not necessarily indicate deficiencies in the ROMS model. The ROMS mean sections are for a 10 year simulation (2002-2011) that is driven by BRAN3 boundaries, while the CARS data is averaged over the period of modern measurement. Furthermore, the CARS resolution is 0.5 degree in the horizontal (~56km) compared to 3-6km for the ROMS grid, so clearly the shelf and shelf break cannot be well resolved by CARS. To be confident that these small differences do not indicate deficiencies in our ROMS model, we have also plotted difference plots comparing BRAN3 from the same 10-year period (2002-2011) and CARS. We have not included the difference plots in the manuscript as the differences have little meaning and are likely to be misleading to the reader. Instead we have included some additional discussion in the text.

"The mean temperature sections compare very well with the corresponding mean temperature sections from BRAN3 over the 10-year period, as expected as the ROMS model receives its boundary forcing from BRAN3. The mean temperature sections also match the corresponding mean temperature sections from the CSIRO Atlas of Regional Seas climatology well (CARS, Ridgway et al. (2002)), shown in the bottom panel of Figure 3. There are some small differences that are not surprising given that the CARS data covers a longer averaging period and is mapped at a much courser horizontal resolution (0.5°). In particular, difference plots (not shown) reveal some differences over the continental shelf and slope, which is not well resolved by CARS. The comparison provides confidence that our ROMS model is representing the mean thermal structure of the ocean well."
Page 6, lines 6-17 of new revision.

*Page 6, lines 22-24: Does the 2 year free run start from the end of the 10 year free run?*

No, the 10-yr free run goes from Jan 1 2002-Dec 30 2011. The 2-yr free run goes from Dec 1 2011-Dec 30 2013. They are independent; the 2-yr run is not a continuation of the 10-yr run. The 2-yr free run is initialised from BRAN, not from the 10-yr free run, and has a 1-month spin up before we begin assimilation on Jan 1 2012. This has been made more clear in the text, "The reanalysis model uses initial conditions and boundary forcing from BRAN3 and atmospheric forcing provided by the 12km resolution BOM ACCESS analysis, which was not available to over the 10yr free run testing period described above. The simulation is spun-up over a 1-month period before we begin assimilation on Jan 1 2012. A surface heat flux correction was applied such that the new atmospheric surface forcing is in balance with SST from BRAN3 for each month."
Page 7, lines 1-4 of new revision.

*Page 7, line 10: Reword as "... deviations OF THE MODEL from the observations..."*
*Page 7, line 11: Reword as "J comprises a term that represents the difference..."*
*Page 7, line 14: Reword as "covariance, and a term that penalizes...."*

These corrections are no longer relevant as this section (Section 3.2) has been completely reworked, as per the comment above.

*Page 7, line 34: "free within the known uncertainties in the system" - this does not make grammatical sense.*

This sentence has been moved to the beginning of the following section (Section 3.3) and reworded as "To do this we solve for the nonlinear ocean solution that is dynamically consistent with the observations and is free within the uncertainties in the system".
Page 9, lines 23-25 of new revision.

*Page 8, line 8: Reword as "...INDICATED that for this MODEL configuration, the linear ...*

This has been reworded as "Linearity experiments (not shown) indicated that for this model configuration, the linear ....".
Page 9, line 34 of new revision.

*Page 8, line 10: Replace "feasible" with "reasonable" or "affordable"*

Replaced with "reasonable".
Page 9, line 31 of new revision.

*Page 9: lines 13, 15 and elsewhere: You imply here, and elsewhere, that the model is capturing submesoscale variability. The submesoscale is generally viewed as being in the 1-10km scale range, so your model will not resolving the submesoscale. You should remove the references to the submesoscale circulation.*

This has been reworded to not reference 'submesoscale' explicitly. At page 11, lines 5-6 of the new revision, we have changed the wording to read "the model resolves far more structure at smaller spatial scales than is capable in the observations.", instead of "the model resolves far more structure in the submesoscale than is capable in the observations."

*Page 9, line 23: Reword as "... observation ERROR VARIANCE for the assimilation is CHOSEN to BE the square..."*

Throughout we have checked consistency with the use of the term 'uncertainty' (which has the same units as the state variable itself ie. degC for temperature) and 'error variance' (which is a squared quantity). At page 11, lines 18-20 of the new revision we write "As the resolution of the data is similar to the resolution of the model, the observation uncertainty for the assimilation is chosen to be equal to this product error. ", instead of "As the resolution of the data is similar to the resolution of the model, the observation uncertainty for the assimilation is set to the square of this product error."

*Page 9, line 29: "not resolved by the model" - I think you mean not resolved by the data*

This has been replaced with "to account for additional uncertainty due to processes not resolved by the observations or the model". We need to include ''the model'' here also as, although the model resolves more structure than the data in this case, it does not resolve all of the processes that affect observed SSS.
Page 11, line 25 of new revision.

*Page 11, line 23: "covariance" misspelled.*

Changed, thanks.
Page 13, line 16 of new revision.

*Page 13, line 17: lower than the model AND OBSERVATION prior*

This sentence has been moved from Section 3.5 to Section 3.3 and now reads "The goal of the assimilation is to combine an uncertain model with uncertain observations to generate a circulation estimate that has reduced uncertainty and better represents the observations.", instead of "The goal of the assimilation is combine an uncertain model with uncertain observations to reduce the analysis uncertainty to be lower than the model prior."
Page 9, lines 22-23 of new revision.

*Page 13, line 19: Reword as "These climatological varinces provide..."*

This has been addressed.
Page 16, line 18 of new revision.
Also note, the entire section (section 3.5) on the model prior uncertainties has been expanded, in response to the general comments by this reviewer requesting a more detailed account of the 4D-Var system and to comments by referee #2.

*Page 14, line 22: You say here that because you have only estimates of the variances, the background covariance of each field is estimated as a diffusion operator. This is not the reason that covariances are estimated this way. I suggest you go back to the original papers on modeling covariances using diffusion equations and brush up on some of the ideas (e.g. balanced versus unbalanced flows, etc).*

This section has been expanded to explain the formulation of P. We have included the sentence "Because of its size P cannot be estimated completely or stored and we estimate P by factorisation, as described in Weaver and Courtier (2001), such that…" for example.
Section 3.5, page 15, line 13 to page 16, line 22 of new revision.

*Page 14, lines 24 and 25: Moore et al (2011) is not appropriate reference - you should refer to the appropriate equations in the orginal paper by Desroziers et al (2005).*

We now refer to the reference by Desroziers et al 2005.
Page 16, line 29 to page 17, line 2 of new revision.

*Page 15, line 4: The authors claim that they have generated a "near-optimum" minimization. This is most certainly not the case since the background and observation error covariances they use are very far from being the true error covariances. The authors should tone this down or remove it.*

We have changed the sentence that summarises the 'Consistency of observation and model uncertainties' section to read "Overall, we have rigorously tuned the assimilation system and are confident that the prior uncertainties are well specified such that the assimilation achieves reduced analysis uncertainty by reduction of the cost function for each assimilation interval."
Page 17, lines 23-25 of new revision.

*Page 15, line 13: Reword as $t_1, t_2, ... t_n$ instead as $t_1-t_2$ since this looks like the time difference.*

This has been changed and the equations are no longer written in the text, so that they are clearer.
Page 18, line 15 of new revision.

*Figure 12 and 13: You show only the complex correlations - what are the rms errors in the current speeds? It would interesting to know this also.*

We have added an additional paragraph in Section 4.3.6 describing the RMS errors in the radial current speeds.
Page 21, lines 14-18 of new revision.

*Page 18, line 18: Say "diagnosed" instead of "posterior" - see specific comment (2) above.*

This has been changed and the sentence now reads "The performance of the reanalysis is dependent on prior assumptions of the model background and observation error co-variances and we show that the prior and diagnosed model and observation uncertainties are consistent."
Page 22, line 22 of new revision.

*Figure 4: The light grey points are almost illegible (they are in legend).*

Figure 4: This figure has been made clearer.
Page 31 of new revision.

*All figure: The fonts are tooo small to legible in many cases (eg Fig. 3, Fig. 8, Fig. 9, Fig. 12). The legends in particular are difficult to read.*

All figures: The font sizes have been increased and all figures made clearer where deemed necessary.
The panel showing the location of the sections from Figure 3 has been removed, and the sections are now shown in Figure 2. This allows Figure 3 to be much clearer.

Referee #2:

Specific Comments:

*Page 2, Line 18: Sentence beginning with: "Much of the effort . . ." This sentence is awkward to read. But more importantly see: O'Kane, T.J., Oke, P.R. and Sandery, P.A., 2011. Predicting the east Australian current. Ocean Modelling, 38(3), pp.251-266. for additional work in this area.*

ThIs additional work has been recognised and referenced.
Page 2, lines 23-24 of new revision.

*Page 2, Line 25: You state that a major advantage of the 4D-Var system is the fact that it relies on a linearised version of the governing equations. It could be contested that this is in fact a disadvantage in highly non-linear systems and the assimilation window needs to be shortened such the the system behaviour is quasi-linear. I suggest you make it clear at this point that there are also some limitations (as all systems have), that need to be accounted for.*

This has been reworded to read "Using the linearised equations allows dynamical connections between state variables to propagate information from observed variables to unobserved, dynamically-linked variables. Because the linearised version of the governing equations is used, rather than the full nonlinear version, the assimilation interval length is limited such that the linear assumption remains reasonably valid and the nonlinearities do not grow too large."
Page 2, lines 30-34 of new revision.

*Page 5, Line 33: "downward tilt", this is not a very intuitive description of the thermocline position. e.g. Does this mean the thermocline is uplifted near the shelf break, or depressed near the shelf break?*

Page 6, Line 7: We have reworded downward to "upslope" tilt.

*Page 7, line 4: In this para you give a high level description of the way IS4D-Var work.
I suggest you add an additional sentence explaining what the increment is. In the previous para you say the observations are used to constrain the model increments, again without introduce the concept of what an increment is. A sentence such as: "The IS4D-Var system generates a vector (time series) of increments that are added to the initial conditions (and forcing), to minimise the cost function."*

Section 3.2 has been reworked to include more detail on the 4D-Var scheme, in response to comments made by Referee #1 and this comment. The 'increment' is now well defined.
Page 7, line 13 to page 9, line 19 of new revision. The increment is defined in Equation 5, page 8 line 8.

*Page 7, line 20: How many inner loops did you run, and what was time series for the "J ratio" or reduction in the cost function?*

The number of inner loops is detailed in the Assimilation Configuration section (Section 3.3) that follows.
Page 9, line 31 of new revision.

A section on the cost function reduction has been added (Section 4.2) to the Reanalysis Evaluation section.
Page 17, line 26 to page 18, line 5 of new revision.

*Page 8, line 16: How is the blended product constructed? Is this done my weighting in the assimilation scheme, or as a post processing step.*

It is a post processing step and has been explained more clearly in the text.
"We overlap the 5-day assimilation windows by one-day, such that the initial conditions for the subsequent assimilation window are 4 days after the start of the current window. The overlap allows us to produce a blended product which is constructed as a post processing step using a weighted average of the overlapping times from adjacent assimilation windows to build a continuous signal."

Page 10, lines 1-5 of new revision.

*Page 8: There is some duplication of material between section 3.2 and section 3.3.*

This comment has been addressed. Both sections were reworked based on comments by Referee #1 and this reviewer.

*Page 8, line 29: A good discussion of representation error can be found in:*
*Oke, P.R. and Sakov, P., 2008. Representation error of oceanic observations for data assimilation. Journal of Atmospheric and Oceanic Technology, 25(6), pp.1004-1017.*

We thank the reviewer for alerting us of this good, clear discussion of representation error, and have included the reference.

Page 10, lines 27-28 of new revision.

*Section 3.4.1: The AVISO product is a daily gridded product that is derived from along track altimetry. Why not use the raw along track altimetry and use the IS4D-Var system constrain the SSH? By using a daily gridded product, some cells will have a higher error due to the statistical interpolation, while others that lie on the satellite track, will have a low uncertainty, therefore, by assuming a 6cm error, you are degrading the along track product to the error estimate used for the interpolated cells. Why not use the IS4D-Var system to assimilate the along track data, then use the model dynamics (which should be better than the statistical model) to constrain the unobserved cells? What is the implication of using the interpolated SSH cells with the observed temp and salinity (if/when this occurs), does J fail to converge?*

From experience (pers. comm. Brian Powell and John Wilkin), assimilating along-track with 4D-Var in a large mesoscale dominated region does not work well. If we apply a single alongtrack with 4DVar, the assimilation scheme can create a surface gravity wave in the barotropic to match the SSH observations, rather than projecting into the baroclinic. This problem occurs because we aren't prescribing the covariance between the SSH and the baroclinic (i.e. the balanced terms in the background error covariance matrix). This is possible to do in ROMS but has not yet been done with much success; something we would like to explore in the future (so that we could use the alongtrack SSH data). For this work we use AVISO, which gives a daily, synoptic view of SSH. Applying a spatial product ensures the SSH is projected into the subsurface. A note has been made in the text that reads "The gridded AVISO product is used to constrain SSH, rather than the along-track altimetry, to ensure that the constraint is projected into the baroclinic ocean state solution. The use of along-track SSH data successfully with 4D-Var relies on the prescription of balanced terms in the background error covariance matrix to describe the covariance between SSH and the subsurface ocean (refer to Section 3.5). This is a topic of further research."

Page 11, lines 10-14 of new revision.

The representation errors applied to the SSH and the T and S observations account for the use of the spatially interpolated SSH product in cells where there are T and S observations.

*Section 3.4.4: The errors that are assigned to Argo profiles appear very large. How can you justify the error on an Argo float being up to a factor of 3 times larger than an SST observation? Depending on the grid resolution of the model, that cannot just be written off as representation error. Do you need to artificially inflate the prior over the floats to allow J to converge due to the gridded AVISO product?*

The nominal minimum observation uncertainty profiles are shown in Figure 7. They were scaled based on preliminary assimilations and checks against the diagnostics described in Section 4.1 (this is explained in Section 3.4.4, page 12 lines 5-8). At the surface for temperature, an uncertainty value of 0.5 degC is applied, compared to 0.4 degC for SST. They are not a factor of 3 times larger than the SST observation errors. The range that was included in the text is misleading, "The uncertainties for Argo range from

0.12-1.2degC (0.06-0.16) for temperature (salt).", as the higher end refers to a small number of cases where the standard deviation of all observations in a single cell reaches up to 1.2 degC. Most of the uncertainties follow the profiles in Figure 7. This unnecessary sentence has been removed to avoid confusion.

*Page 13, Line 23: Is it usual in 4D-Var system to use a diffusion operator as a prior?*
*Is this not the same as using an isotropic 3D Gaussian function? Is there a reference that can support your use of the diffusion operator?*

Section 3.5 on model prior uncertainties has been reworked to explain the formulation of the background error covariance matrix in more detail. It is usual in 4D-Var systems to use a diffusion operator as a prior. This is explained in Weaver and Courtier 2001 (reference added at page 15, line 15 of new revision).

The solution of a diffusion equation is a Gaussian distribution. We are assuming the matrix of background error correlations is a Gaussian Spatial Correlation function and solving for it using the diffusion operator. Refer to page 15, lines 21-22.

*Section 3.5 What is the implication of choosing sub-optimal de-correlation length scales for the reanalysis system.*

In Section 3.5 we note that, although we assume the characteristic length scales of variability to be homogenous and isotropic, this is not likely to be the case in this region. In particular, the horizontal length scales of variability will differ from the shelf to the offshore Tasman Sea region. See page 15, lines 28-30.

It is noted that further work towards specifying anisotropic length scales would be warranted. On page 16, lines 13-16, we note that "The background error covariance matrix plays an important role in determining the spatial structure of the analysis increment and, in this oceanic region, the horizontal and vertical scales of variability differ between the mesoscale eddy field in the Tasman Sea and the smaller-scale shelf processes. Further research on the impact of applying anisotropic correlation length scales on system performance is warranted."

*Section 4.2 Are the observations used to calculate the RMSD'd in this section withheld from the assimilation system, or are you reporting the forecast error statistics (i.e. before assimilation takes place)? Is the RMSD reduction between the free-run and assimilating run due to a reduction in bias or due to the dynamics features being correctly predicted?*

This is now Section 4.3 in the new revision. The observations used here are the assimilated observations, so we are reporting the analysis error statistics. We do not report forecast errors as this was decided to be outside of the scope of this paper, which focuses on reanalysis evaluation (not forecast skill). This has been made clearer by renaming the section title as "4.3 Reanalysis Comparison to Assimilated Observations". Section 4.4 is titled "Reanalysis Comparison to Independent Observations".

The RMSD reduction between the free-run and assimilating run is predominantly due to the dynamical features being correctly predicted. The bias between the free run and the observations was small. This has been clarified in the text,

For SST: "The free-running model and SST observations exhibit no net bias over the 2 years (as mentioned in Section 3.1), indicating that the the RMSD reduction between the free-run and the analysis is due to improved prediction of the dynamical features rather than a reduction in bias.", Page 19, lines 10-12 of new revision.

For Argo, XBT and gliders: "To investigate the relative contribution of improved representation of dynamical features and reduction in bias to the RMSD reduction between the free-run and the analysis, we also compute the RMSD between the free run and the 'bias adjusted observations'. The 'bias adjusted observations' have the bias between the observations and the free run removed and, for each depth bin, are given by $v_o(t) - (\bar{v}_o - \bar{v}_f)$, where $v_o$ are the observations in the depth bin for observation times $t = t_1, t_2, ....t_n$, $v_f$ are the corresponding values from the free-run and the overbar represents the time-mean of the variables over the 2-year period. For the Argo and XBT observations, the bias between the free-run and the observations is small and the RMSD$_{Freerun-Obs}$ and the RMSD between the free run and the 'bias adjusted observations' (blue and grey dashed lines in Figure 11, respectively) match closely. The RMSD reduction for the analysis (magenta line) is due to better representation of the dynamical features. The vast majority of glider observations are taken on the continental shelf in water depths less than 100m. For these shallow observations, the bias between the free-run and the observations is small, the RMSD$_{Freerun-Obs}$ and the RMSD between the free run and the 'bias adjusted observations' match closely, and the RMSD reduction in the analysis represents improved representation of the dynamical features. The glider observations below 100m represent only 2 separate glider missions (refer to Section 3.4.9), so the bias has little meaning."
Page 20, lines 1-16, and added line of Figure 11

*General Comments:*

*The second half of the introduction reads like a combination of the abstract, methods and results/conclusion section. To avoid replication of information, it would be worthwhile considering moving the material from page 3, line 3 - line 25 into other more appropriate sections.*

The purpose of the second half of the introduction is to summarise the methods, results and conclusions. We consider it valuable to introduce the 4D-Var scheme here. We have checked for repetition and removed some text appropriately.

*Can the the inner loops of the assimilation scheme generate unrealistic initial conditions or forcing data?*

Because 4D-Var uses the (linearised) primitive equations to constrain the adjustments of the model, within the inner loops everything is - by definition - self-consistent. The action of the tangent linear and adjoint models creates the dynamical covariance such that any perturbation is consistent with the dynamics of the ocean. As such, the adjusted initial conditions are always consistent with the (linearised) model dynamics.

For the atmospheric perturbations, we don't have an atmospheric model to constrain adjustments, but the ocean response to heat, salt, and wind fluxes tightly constrain what the adjustments to the atmosphere can be made. So, in a technical sense, we could create an atmosphere that - in itself - is unrealistic, but from the ocean perspective it acts consistently with the ocean dynamics.
This has been included in the text at page 9, lines 7-8 with the sentence "Because 4D-Var uses the (linearised) model equations to constrain the increments, any adjustments to the model are consistent with the dynamics of the ocean".

Biases in the increments can indicate potential errors in the forcing product (or the model configuration) i.e. if we are consistently adding less surface heating over a certain area and/or cooling the initial conditions, is the atmospheric forcing consistently over estimating heating there? or is the ocean model not resolving a process that is uplifting cool water from depth? For this reanalysis, we find no significant bias is any of the increments. This is mentioned at Page 10, line 7

*Is it great to see the Aquarius data being used, can you give any indication as to the value of this data in the assimilation scheme? i.e. how much does it contribute to the reduction in J?*

The Aquarius salinity data was included but for this assimilation configuration provides little constraint. An additional paragraph has been added to the report, Section 4.3.3, page 19, lines 15-20 of new revision.

"The Aquarius SSS data was included but for this assimilation configuration provides little constraint. RMSObsAnom for SSS is 0.15-0.3 over most of the model domain (up to 0.5 at a few points close to the coast). The Aquarius product error itself is 0.2 and our specified observation error is 0.4, which is greater than the typical variability in SSS over most of the domain, so the assimilation does little to match the SSS observations as they are so uncertain. The RMSDFreerun–Obs, RMSDAnalysis–Obs and RMSObsAnom are all of similar magnitude. Subsurface salinity dominates the salinity cost function as the prescribed observation uncertainties are considerably higher for SSS than the uncertainties specified for the in-situ salinity observations."

*The authors should be commended in their use of such a wide variety of observational data to constrain the reanalysis. It would be fascinating to know the relative impacts of each of the observation platforms? This study presents a detailed assessment of the reanalysis skill, but I was left wondering about the following points. These may be outside of the scope of*

*this study, but I thought I'd pass them on as food for thought:*
*1.) The 10 year control run was assessed against the climatological EAC transports, and shown to give a good statistical representation of the mean transport. What was the difference in transport between the 2 year free run, and the reanalaysis. Is there a substantial difference between the estimated transports? Which observations had the most impact on quantifying the transport (e.g. see the study of Moore et al., 2011)? Such a study may provide insight into how an observing system should be configured to monitor and observe the impacts of Climate change on the EAC strength.*

This is the subject of our new work. However we need to publish the model configuration and validation as a first step (ie this paper), in order to do further studies. In our new work we have quantified the relative impacts of each of the observation platforms on estimates of transport through 3 cross sections along the east coast (at the EAC separation zone, upstream, and downstream – the same sections as Figure 3 of this paper). The transport (averaged over the 5-day windows) differences between background and analysis are of the order of +/- 8 Sv. We look forward to publishing this work soon. In the concluding section we state that this will be future work (Page 23, lines 4-8 of new revision.)

*2.) What value does the high resolution reanalysis give compared to BRAN 3p5?*
*Obviously, the high res reanalysis is needed for nesting purposes, but do the error statistics look different between the course global reanalysis and the high resolution regional reanalysis?*

Also a comparison that we think is very important to make i.e. is the higher resolution giving improved predictive skill? We have set up a 2 year assimilation similar to the one in this study that assimilates only the observations used in BRAN 3p5. We hope to compare both this new ROMS 4D-Var assimilation and BRAN to the independent observation platforms and their forecasts to all observation platforms (assimilated and independent). We hope to secure funding to allow us to complete this work in the near future.
(Note that we assimilate AVISO rather than along-track, which is used in BRAN. We would also like to look into including balanced terms in the background error covariance matrix to allow effective assimilation of along-track data in 4D-Var – see response to Section 3.4.1 comment above)

---

## Author Response (AR2)

In this document we do two things:

- 1. Respond to reviewers' comments and suggestions for minor revisions, as received on Jul 12 2016.
- 2. Identify improvements to the reanalysis results after rerunning the simulation. After the paper was submitted we identified there was an error in the wind forcing for approximately half of the 2-year period. The atmospheric forcing was corrected, and the model was re run. We present the improvements here. As expected, with the corrected forcing we find only small changes to the results, which are not significant to the overall presentation of the reanalysis performance. There are no significant changes to the discussion and conclusions.

**1. Changes made in response to reviewers' comments:**

*Reviewer 1 report on revised manuscript:*

The authors have done a good job addressing my initial comments and concerns. I recommend publication subject to the following minor revisions:

**(1) p3, line: "sensitivity of the ocean circulation" - sensitivity to what?**

We have replaced the phrase "and the minimistaion process can be used to understand the sensitivity of the ocean circulation" with "and the minimistaion process can be used to understand the sensitivity of the modelled ocean circulation to initial conditions, boundary and surface forcing, and model parameters".

(2) p4, line 3 and 4: This sentence is still awkward. How about something like "We use ROMS to simulate the ocean circulation off the south eastern coast of Australia."

We have replaced the awkward sentence "We use the Regional Ocean Modeling System (ROMS, version 3.4) to simulate the atmospherically-forced eddying ocean circulation in the south-eastern Australia oceanic region." with "We use the Regional Ocean Modeling System (ROMS, version 3.4) to simulate the atmospherically-forced eddying ocean circulation off the south eastern coast of Australia."

(3) p7, line 18: Reword as "...variational calculus to solve for increments in model..."

This has been changed accordingly.

**(4) p7, line 21: "normalized deviations" - of what?**

We have replaced the sentence "This is achieved by minimising an objective cost function, \$J\$, that measures normalised deviations from the observations as well as from the modelled background state (the model prior)." With "This is achieved by minimising an objective cost function, \$J\$, that measures normalised deviations of the

modelled ocean state from the observations as well as from the modelled background state (the model prior)."

(5) Equations 1 and 2: In equation (1), t\_{i-1} and t\_i are used to denote a general time interval, while in equation (2) and beyond the notation is changed to t\_0 and t\_i. The use of t\_i for two different times is confusing - this should be fixed. I assume that t\_0 is the initial time for each data assimilation cycle? This should be stated in the text.

This has been corrected. We thank the reviewer for this comment.

(6) p9, lines 7-9: You imply in the text, and in your reply to the other reviewer, that because 4D-Var employs the model equations to constrain the circulation increments they are "consistent with the dynamics of the circulation." This will not generally be true for the circulation estimate at initial time, t\_0. Unless dynamical balance (eg quasi-geostrophic balance) is explicitly imposed as a constraint at t\_0, the initial condition increments can be quite unbalanced, leading to subsequent initialisation shocks and gravity wave generation. You should clarify this statement, or remove it.

The increments are constrained to be a solution of the (TL) model equations. This does not imply dynamical balance of the initial conditions. We have removed this phrase.

(7) p10, line 1: Reword as "THOSE 4 days after".

It does not make sense to reword this phrase as so. We have clarified what we mean here by replacing, "We overlap the 5-day assimilation windows by one-day, such that the initial conditions for the subsequent assimilation window are 4 days after the start of the current window." with "We overlap the 5-day assimilation windows by one-day, such that each subsequent assimilation cycle is initialised 4 days after the start of the previous 5-day cycle."

(8) p17, line 21: It is stated here that gamma>1 represents an under-estimate of the error covariances, while gamma

Figure 4. Temperature-Salinity diagram for the Argo observations and corresponding values from the 2yr free run for 2012-2013.

Updated version:

Figure 4. Temperature-Salinity diagram for the Argo observations and corresponding values from the 2yr free run for 2012-2013.

Page 17, line 7 SSH diagnosed errors, 4.1-8.6cm with a mean of 5.7cm changed to 4.1-8.4cm with a mean of 5.8cm line 11 Subsurface temperature diagnosed errors, 0.50C changed to 0.48C line 15 Radial diagnosed errors, 12m/s changed to 11m/s lines 22-23 Optimality range from 0.44-1.66 with a mean of 0.84 changed to 0.43-1.72 with a mean of 0.81

Page 17, last line, page 18, line 1 NLM J reduction changed from 54% to 52% Figure 8 has been updated, but the changes are not significant

Previous version: